# Onset of taste bud cell renewal starts at birth and coincides with a shift in SHH function

Erin J Golden[1,2], Eric D Larson[2,3], Lauren A Shechtman[1,2], G Devon Trahan[4], Dany Gaillard[1,2], Timothy J Fellin[1,2], Jennifer K Scott[1,2], Kenneth L Jones[4], Linda A Barlow[1,2]*

[1]Department of Cell & Developmental Biology, University of Colorado Anschutz Medical Campus, Aurora, United States; [2]The Rocky Mountain Taste and Smell Center, University of Colorado Anschutz Medical Campus, Aurora, United States; [3]Department of Otolaryngology, University of Colorado Anschutz Medical Campus, Aurora, United States; [4]Department of Pediatrics, Section of Hematology, Oncology, and Bone Marrow Transplant, University of Colorado Anschutz Medical Campus, Aurora, United States

**Abstract** Embryonic taste bud primordia are specified as taste placodes on the tongue surface and differentiate into the first taste receptor cells (TRCs) at birth. Throughout adult life, TRCs are continually regenerated from epithelial progenitors. Sonic hedgehog (SHH) signaling regulates TRC development and renewal, repressing taste fate embryonically, but promoting TRC differentiation in adults. Here, using mouse models, we show TRC renewal initiates at birth and coincides with onset of SHHs pro-taste function. Using transcriptional profiling to explore molecular regulators of renewal, we identified *Foxa1* and *Foxa2* as potential SHH target genes in lingual progenitors at birth and show that SHH overexpression in vivo alters FoxA1 and FoxA2 expression relevant to taste buds. We further bioinformatically identify genes relevant to cell adhesion and cell locomotion likely regulated by FOXA1;FOXA2 and show that expression of these candidates is also altered by forced SHH expression. We present a new model where SHH promotes TRC differentiation by regulating changes in epithelial cell adhesion and migration.

*For correspondence:
LINDA.BARLOW@CUANSCHUTZ.
EDU

**Competing interests:** The authors declare that no competing interests exist.

## Introduction

Taste buds are the primary end organs of the gustatory system, which reside in three types of specialized epithelial papillae on the tongue: fungiform (FFP) in the anterior tongue and circumvallate and foliate, posteriorly. Regardless of papilla location, each taste bud houses a collection of heterogeneous taste receptor cells (TRCs) that transduce taste stimuli, including sweet, umami, salt, sour, and bitter, to signal palatability, nutritional value, and/or danger of substances in the oral cavity. These signals are conveyed from taste buds to the brain via gustatory nerve fibers of the VIIth and IXth cranial nerves. Despite some neuronal characteristics that accompany their sensory function, all TRCs are modified epithelial cells and are continuously renewed (see *Barlow and Klein, 2015*). Adult taste cells are generated from progenitors adjacent to taste buds that express cytokeratin 14 (KRT14) and KRT5 (*Gaillard et al., 2015*; *Okubo et al., 2009*), a population of basal keratinocytes that also gives rise to the non-taste lingual epithelium that covers the tongue surface. Taste-fated daughter cells exit the cell cycle, enter buds as post-mitotic taste precursor cells that express sonic hedgehog (SHH), and differentiate directly into functional TRCs (*Miura et al., 2006*; *Miura et al., 2014*).

Embryonically, taste bud primordia first develop as focal epithelial thickenings, or taste placodes, on the mouse tongue at mid-gestation (embryonic day [E] 12.0) (*Mistretta and Bosma, 1972*; *Mistretta and Liu, 2006*). In the anterior tongue, placodes undergo morphogenesis to form FFP, and taste bud primordia are first innervated by gustatory fibers at E14.5 (*Lopez and Krimm, 2006*); differentiated TRCs are observed in the first postnatal week (*Ohtubo et al., 2012*; *Zhang et al., 2008*). Shh and target genes *Gli1* and *Ptch1* are initially co-expressed broadly by lingual epithelium; at E12.5, this pattern resolves to placode-specific SHH expression as taste placodes are specified, while placode-adjacent epithelial cells remain *Gli1+/Ptch1+* (*Hall et al., 1999*; *Jung et al., 1999*). Embryonic lineage tracing reveals SHH+ placode cells become the first differentiated TRCs after birth (*Thirumangalathu et al., 2009*). However, SHH+ placodes do not give rise to adult taste progenitors, as SHH-derived TRCs are steadily lost from taste buds within a few postnatal months (*Thirumangalathu et al., 2009*). By contrast, lineage tracing of KRT14+ cells initiated in the first postnatal week labels small numbers of taste cells, suggesting that the adult progenitor population is activated in the days following birth (*Okubo et al., 2009*). It remains to be clarified as to when and how KRT14+ progenitors activate and begin to contribute TRCs to maintain taste buds.

In addition to marking taste placodes in embryonic tongue and post-mitotic precursor cells in mature taste buds, SHH is a key regulator of taste bud development and homeostasis. In embryos, SHH functions to repress taste fate as taste placodes are specified and patterned (*El Shahawy et al., 2017*; *Hall et al., 2003*; *Iwatsuki et al., 2007*; *Mistretta et al., 2003*); while once taste placodes are established, embryonic taste primordia no longer respond to pharmacological manipulation of hedgehog signaling (*Liu et al., 2004*). By contrast, in adults, SHH promotes and is required for TRC differentiation. Specifically, ectopic expression of SHH drives formation of ectopic taste buds (*Castillo et al., 2014*), while pharmacological inhibition or genetic deletion of Hedgehog (Hh) pathway components leads to loss of taste buds (*Castillo-Azofeifa et al., 2017*; *Ermilov et al., 2016*; *Kumari et al., 2015*). Whether the shift in SHH function coincides with progenitor activation has not been explored. SOX2, an SRY-related HMG-box transcription factor, is also a key regulator of embryonic taste bud formation and adult taste cell renewal, and in both contexts, SOX2 is required for TRC differentiation (*Castillo-Azofeifa et al., 2018*; *Ohmoto et al., 2020*; *Okubo et al., 2006*). Additionally, SOX2 function is required downstream of SHH, at least in adult taste cell homeostasis.

Here, we sought to define when TRC renewal from adult progenitors occurs, if this renewal coincides with functional shifts in the response of lingual epithelium to SHH, and identify genetic components, in addition to SOX2, that may function downstream of SHH in TRC renewal.

## Results

### Once specified, taste placodes do not receive additional cells from proliferative KRT14+ basal keratinocytes during embryogenesis

While KRT8 and KRT14 are well known to be expressed embryonically by taste placodes and taste bud primordia, and developing lingual progenitors, respectively (*Iwasaki et al., 2003*; *Mbiene and Roberts, 2003*), their contemporaneous expression had not been carefully described for the developing anterior lingual epithelium. Thus, we performed immunofluorescence for KRT8 and KRT14 on sections of embryonic tongues at progressive stages (*Figure 1*). Before taste placodes emerge, the tongue anlage is covered by a simple bilayered epithelium that broadly expresses KRT8 (*Mbiene and Roberts, 2003*). We observed KRT14 co-expression with KRT8 throughout the lingual epithelium at E12.0 (*Figure 1A–C'*). At E13.5, following placode specification, KRT8 is expressed in placodes and downregulated in non-taste epithelium, although, consistent with previous reports, KRT8 persists in superficial periderm (*Figure 1D–F*, yellow arrows in E) (*Mbiene and Roberts, 2003*). At E13.5, KRT14 expression is mostly lost from placodes but evident in cells adjacent to placodes as well as throughout the lingual epithelium. These patterns are refined at E16.5; in each FFP, KRT8+ taste bud primordia are KRT14− immunonegative (*Figure 1I*, asterisk), and KRT8 expression is fully absent from KRT14+ non-taste lingual epithelium as the periderm is lost by this stage (*Sengel, 1976*; *Figure 1G,H*). Thus, taste placodes emerge from embryonic stratifying epithelium that co-expresses KRT14 and KRT8; placode specification results in downregulation of placodal KRT14 and maintenance of KRT8 in taste bud primordia, while KRT14+ non-taste epithelium downregulates KRT8.

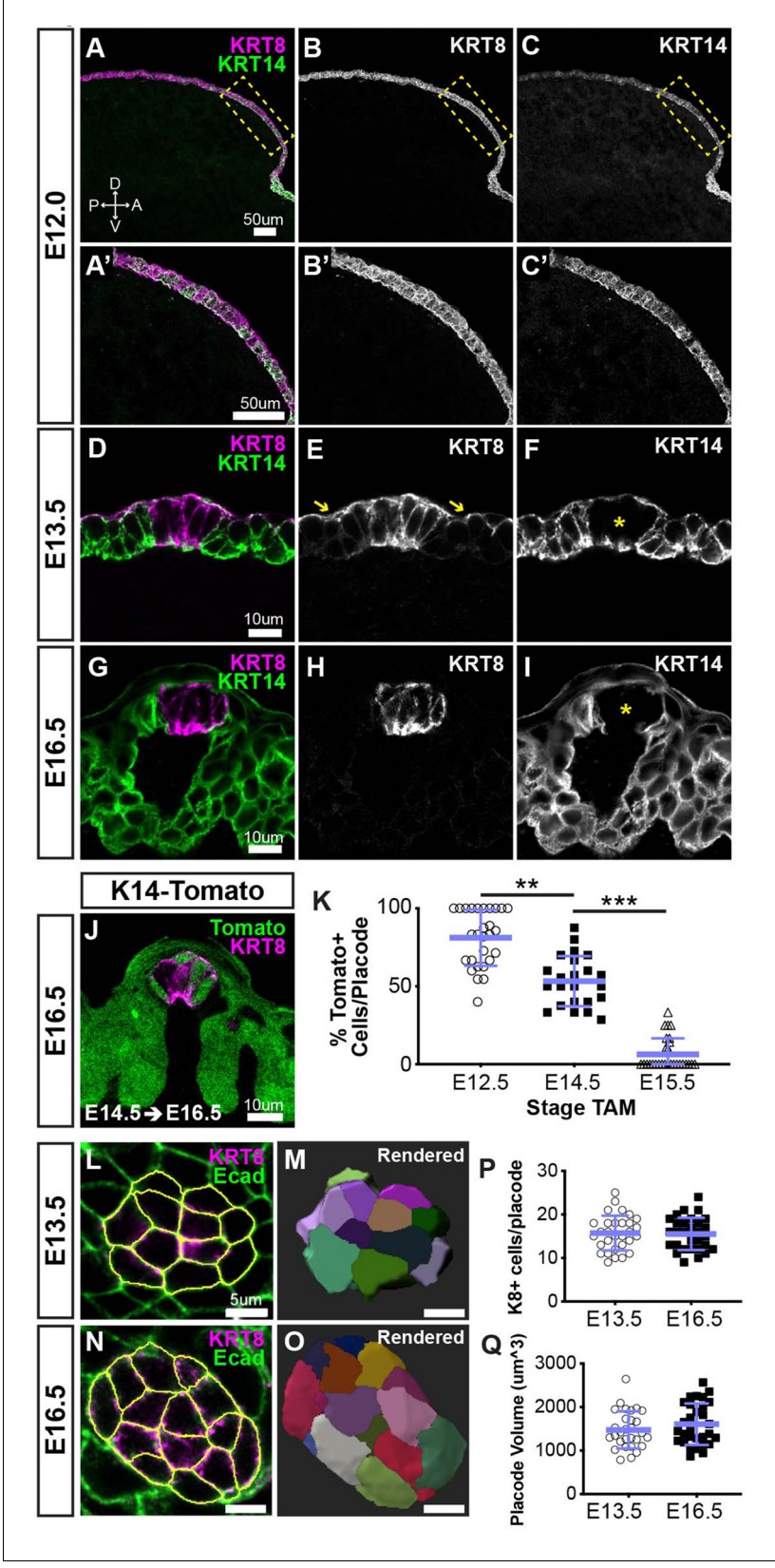

**Figure 1.** Taste placodes arise from KRT14+ progenitors that do not contribute further to taste bud primordia during embryogenesis. (**A–C'**) Before taste placodes are evident (E12.0), KRT14 (green) and KRT8 (magenta) are co-expressed in lingual epithelium. (**D–F**) At E13.5, KRT8 expression is expressed by taste placodes (asterisk in **F**), *Figure 1 continued on next page*

*Figure 1 continued*

and surface periderm (**E**, yellow arrows). KRT14 is evident only basally and apically in placodes (asterisk in **F**) but remains well expressed in non-taste epithelium. (**G–I**) At E16.5, KRT8 is expressed exclusively by taste bud primordia (**H**), which lack KRT14, while non-taste basal epithelial cells are robustly KRT14+. (**J**) *Krt14^CreERT2^; R26R^tdTomato^* (KRT14-tomato; green) induced at E12.5 labels a subset of KRT8+ (magenta) placode cells at E14.5. (**A–J** are confocal optical sections acquired at 0.75 μm.) (**K**) Quantification of placodal KRT14-Tomato+ cells 48 hr after tamoxifen induction at progressive stages, for example tamoxifen at E12.5, analysis at E14.5. Blue bars: mean ± SD (n = 3 animals per stage, 6–11 placodes per animal, open and shaded shapes). Student's t-test **p<0.01, ***p<0.0001. (**L–O**) E13.5 and E16.5 taste placodes (KRT8+, magenta) in E-cadherin (Ecad) immunostained whole tongues were imaged through their apical-to-basal extent, 3D reconstructed and individual cells defined in Imaris (see Materials and methods). White outlines (**L, N**) and randomly assigned colors (**M, O**) indicate individual placode cells. (**P, Q**) Total cell number and placode volume did not differ between stages. Blue bars: mean ± SD (n = 3 animals per stage, 10 placodes per animal, open and shaded shapes).

In adult tongues, KRT8 is a marker of differentiated TRCs (*Knapp et al., 1995*; *Okubo et al., 2009*), which continuously renew from KRT14+ progenitors. Thus, we next asked if KRT14+ cells likewise contribute cells to undifferentiated KRT8+ taste placodes and/or taste bud primordia in embryos. Genetic lineage tracing in *Krt14^CreERT2^;R26R^tdTomato^* (K14-tomato) embryos was activated via tamoxifen at E12.5, E14.5, or E15.5, spanning the period over which taste placodes are patterned and specified. Embryos were harvested for KRT8 immunostaining after 48 hr. At all stages, tomato+ cells were observed in lingual epithelium (*Figure 1J*, and data not shown), consistent with KRT14+ cell generation of lingual epithelium during embryogenesis (*Liu et al., 2007*). No tomato expression was observed in untreated transgenic embryos (not shown). When K14-tomato was induced at E12.5 and assayed 48 hr later (E14.5), most KRT8+ placodal cells were tomato+ (*Figure 1K*). Despite limited KRT14 immunoreactivity in taste placodes at E13.5 (see *Figure 1F*), KRT14-tomato lineage trace initiated at E14.5 resulted in 50% tomato+/KRT8+ cells 48 hr later (E16.5. *Figure 1J,K*). At E17.5 after lineage trace induction at E15.5, most KRT8+ taste bud primordia fully lacked tomato+ cells (*Figure 1K*).

Because taste placode cells are largely KRT14-immunonegative at E13.5 (*Figure 1D–F*), we reasoned that tomato-labeling at E16.5 from lineage trace induced at E14.5 (*Figure 1J,K*) could reflect continued addition of new KRT14+ progenitor-derived cells post-placode specification. Taste placodes are post-mitotic (*Farbman and Mbiene, 1991*; *Mbiene and Roberts, 2003*; *Thirumangalathu and Barlow, 2015*); thus, if embryonic KRT14+ progenitors contribute new cells to placodes, the number of cells per placode should increase during development. To determine whether cells were added to taste bud primordia after specification, we tallied the cell number and volume of taste placodes and bud primordia at E13.5 and E16.5, respectively (*Figure 1L–O* and see Materials and methods). We found developing taste buds were static in both cell number and volume (*Figure 1P,Q*), ruling out an embryonic KRT14+ progenitor contribution to taste bud primordia following placode specification. We suspect that low-level KRT14 and/or Cre recombinase expression prolonged lineage trace by tamoxifen induction at E14.5.

## KRT14+ progenitors contribute to taste buds at birth

To define the onset of new taste cell production, we first compared the number of KRT8+ cells per taste bud section at E16.5 and early postnatal day (P2, P9) stages. KRT8+ taste cell number was comparable between E16.5 and P2 but increased significantly between P2 and P9 (*Figure 2A*), suggestive of the addition of new taste cells during the first postnatal week. To track the entry of cells into buds, we first used thymidine analog birthdating at progressive postnatal timepoints. In pups treated with 5-ethynyl-2′-deoxyuridine (EdU) at P0, EdU+ cells were observed within KRT8+ taste buds 48 hr later, indicating the addition of new taste cells begins at birth (*Figure 2B,C*). Comparable rates of intragemmal labeling were observed in postnatal taste buds when EdU was administered at P7 or P14 and quantified at P9 or P16, respectively (*Figure 2B,D,E*). Previous reports indicated that KRT14 lineage tracing initiated at P2 resulted in sparse labeling of taste bud cells at P9 (*Okubo et al., 2009*). To confirm new postnatal taste cells were derived from KRT14+ progenitors, we induced Cre recombination in *Krt14^CreERT2^; R26R^YFP^* (K14-YFP) pups at P0, P7, or P14 and analyzed YFP+ cell distribution 48 hr later (P2, P9, P16). YFP+KRT8+ cells were readily observed within

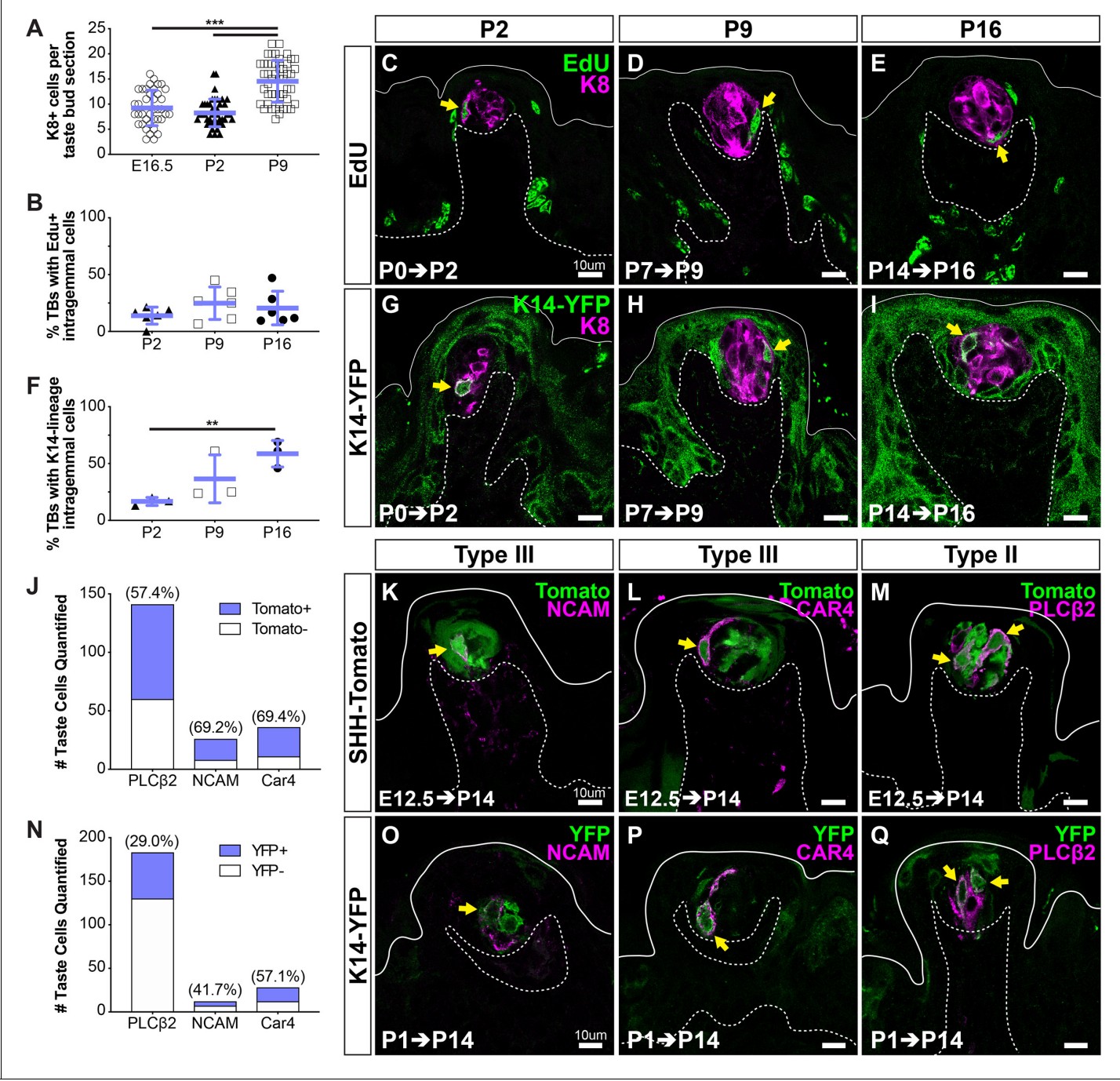

**Figure 2.** KRT14+ progenitor contribution of new cells to taste buds begins at birth. (**A**) Quantification of KRT8+ cells per FFP section reveals taste bud cell number does not increase until P9. Blue bars: mean ± SD (n = 3 animals per stage, 8–18 taste buds per animal, open and shaded shapes) Student's t-test ***p<0.001. (**B–E**) In pups that received EdU at P0, P7, or P9, analysis at 48 hr revealed comparable proportions of taste buds housed newly generated EdU+/KRT8+ cells (yellow arrows in **C–E**) regardless of postnatal day of labeling (EdU green, KRT8 magenta). (**F–I**) Lineage tracing with *Krt14^CreERT2^; R26R^YFP^* (KRT14-YFP) initiated at P0, P7, or P14 assessed at 48 hr showed extensive YFP expression (green) in FFP non-taste epithelium as well as YFP+/KRT8+ cells in taste buds (magenta, yellow arrows in **G–I**). (**C–E, G–I**) Dashed lines delimit the basement membrane; solid lines delimit the epithelial surface. (**B, F**) Blue bars: mean ± SD Student's t-test **p<0.005 (**B**: n = 6 animals per stage, 14–28 taste buds per animal; **F**: n = 3 animals per stage, 10–24 taste buds per animal). (**J–M**) SHH+ taste precursor cells are not lineage restricted. *Shh^CreERT2^; R26R^tdTomato^* (SHH-Tomato, green) mice treated with TAM at E12.5 reveals similar proportions of type III (NCAM+, CAR4+ magenta in **K, L**) and type II (PLCß2+ magenta in **M**) taste cells are tomato+ (green) (N = 3 mice, counts from 24 NCAM+, 30 CAR4+ and 30 PLCß2+ total TBs). (**N–Q**) Postnatally activated KRT14+ progenitors are not lineage restricted. *Krt14^CreERT2^;R26R^YFP^* (K14-YFP, green) mice treated with TAM at P1 labels both type III (NCAM+, CAR4+ magenta in **O, P**) and type II

*Figure 2 continued on next page*

*Figure 2 continued*

(PLCß2 magenta in **Q**). Double labeled cells indicated with yellow arrows in all image panels. Dashed lines delimit the basement membrane; solid lines delimit the epithelial surface.

taste buds at each stage (*Figure 2F–I*); additionally, the proportion of taste buds with KRT14-lineage traced cells increased significantly over the first two postnatal weeks (*Figure 2F–I*). Taken together, our results indicate that KRT14+ progenitors generate new taste cells at birth, and this contribution steadily increases postnatally, leading to taste bud growth.

## Embryonic SHH+ taste placodes and postnatal KRT14+ progenitors give rise to comparable proportions of differentiated taste cell types

Murine taste buds each house ~60 functionally and morphologically heterogeneous cells, including type I support cells, type II detectors of sweet, bitter or umami, and type III sour receptor cells (*Roper and Chaudhari, 2017*). In adult taste buds, SHH+ cells are immediate precursors for all mature TRC types (*Miura et al., 2014*; *Takeda et al., 2013*). Whether SHH+ embryonic taste precursors are similarly competent has not been fully defined. Previously, we used a low-efficiency Cre reporter to trace SHH-expressing taste placode cells from E12.5 into adulthood ($Shh^{CreERT2}$;$R26R^{LacZ}$) (*Thirumangalathu et al., 2009*). At 6 weeks, embryonically derived type I and type II TRCs were readily observed, but lineage labeled type III cells were not, suggesting embryonic taste precursors may be lineage restricted. However, type III cells account for <10% of adult TRCs (*Ma et al., 2007*; *Ohtubo and Yoshii, 2011*), and our previous experimental parameters were not optimized to detect this less common cell population.

Here, to determine if type III, like type I and II TRCs, arise from SHH+ placodes, we employed a high-efficiency Cre reporter allele, $R26R^{tdTomato}$ (*Madisen et al., 2010*) to assess taste cell fate in $Shh^{CreERT2}$; $R26R^{tdTomato}$ (Shh-tomato) pups. Cre induction at E12.5 resulted in robust lineage labeling at P14: 95.2% (±0.01% s.e.m., N = 3 animals) of taste buds were tomato+ with 10.6 (±1.1 s.e.m.) tomato+ cells per taste bud profile (compare with our previous finding of 49.7 ± 7.83% of labeled taste buds with 2.9 ± 0.41 placode-descendent cells per taste bud profile [*Thirumangalathu et al., 2009*]).

Immunostaining for type III TRC markers NCAM and CAR4 revealed significant double labeling in Shh-tomato taste buds. Specifically, ~70% of both NCAM+ cells and CAR4+ cells were tomato+ at P14 (*Figure 2J–L*). Similarly, ~60% of PLCß2+ type II cells were tomato+ at P14, following placodal lineage trace initiated at E12.5 (*Figure 2J,M*). Together with our previously published findings (*Thirumangalathu et al., 2009*), we show SHH+ taste placode cells give rise to all TRC types postnatally. Furthermore, we find placodally derived cells comprise slight majorities of type II and III TRCs at P14, suggesting that unlabeled TRCs were new cells derived from KRT14+ progenitors in the first postnatal weeks.

To test this, we induced K14-YFP lineage trace at P0, P7, or P14, and following a 48 hr or 72 hr chase, immunostained tongue sections for markers of type II and III TRCs as above. However, despite ample numbers of YFP+ cells within taste buds, we did not detect YFP+TRCs immunopositive for PLCß2, NCAM, or CAR4 (data not shown). In adult rodents, type II and III TRCs require ~3.5 days from their terminal division to turn on expression of cell type-specific immunomarkers (*Cho et al., 1998*; *Hamamichi et al., 2006*; *Perea-Martinez et al., 2013*); the lack of double labeling in short -term postnatal lineage tracing here suggested the rate of differentiation of postnatally derived TRCs is comparable to that of adults. Thus, we next examined the fate of newly generated TRCs in tongues from mice traced by induction of K14-YFP at P1 and harvested at P15. Similar to $Shh^{CreERT2}$ lineage tracing, KRT14 lineage tracing resulted in labeling of both type II and III cells in comparable but not identical proportions (*Figure 2N–Q*), indicating that SHH+ taste primordia and postnatal KRT14+ progenitors, nonetheless, have no bias in terms of taste cell type production and are both competent to differentiate the full taste bud lineage.

## SHH represses taste fate during placode specification and promotes taste fate postnatally

The Hh pathway has opposing roles in embryonic versus adult taste epithelium – restricting taste fate during placode specification and promoting taste cell differentiation during adult renewal (*Barlow, 2015*). We next used genetic deletion and activation of the Shh pathway to determine when this shift occurs.

Pharmacologic inhibition or genetic deletion of SHH signaling during placode specification results in overproduction of enlarged taste buds (*El Shahawy et al., 2017*; *Hall et al., 2003*; *Iwatsuki et al., 2007*; *Liu et al., 2004*; *Mistretta et al., 2003*). However, the effects of this disruption on later aspects of taste bud development, including TRC differentiation, have not been assessed. Previously, we observed differentiated type I and II TRCs within a subset of taste buds at E18.5 (*Thirumangalathu and Barlow, 2015*), and further analysis here reveals differentiation of all three TRC types is underway at E17.5 (*Figure 3—figure supplement 1*). To determine whether loss of Shh signaling affects TRC differentiation, we induced *Shh* deletion in *Shh^CreERT2/fl* (Shh-ShhcKO) mice (*Castillo-Azofeifa et al., 2017*; *El Shahawy et al., 2017*). Tamoxifen induction at E13.5 resulted in reduced *Shh* mRNA in E18.5 lingual epithelium, as well as reduced expression of its

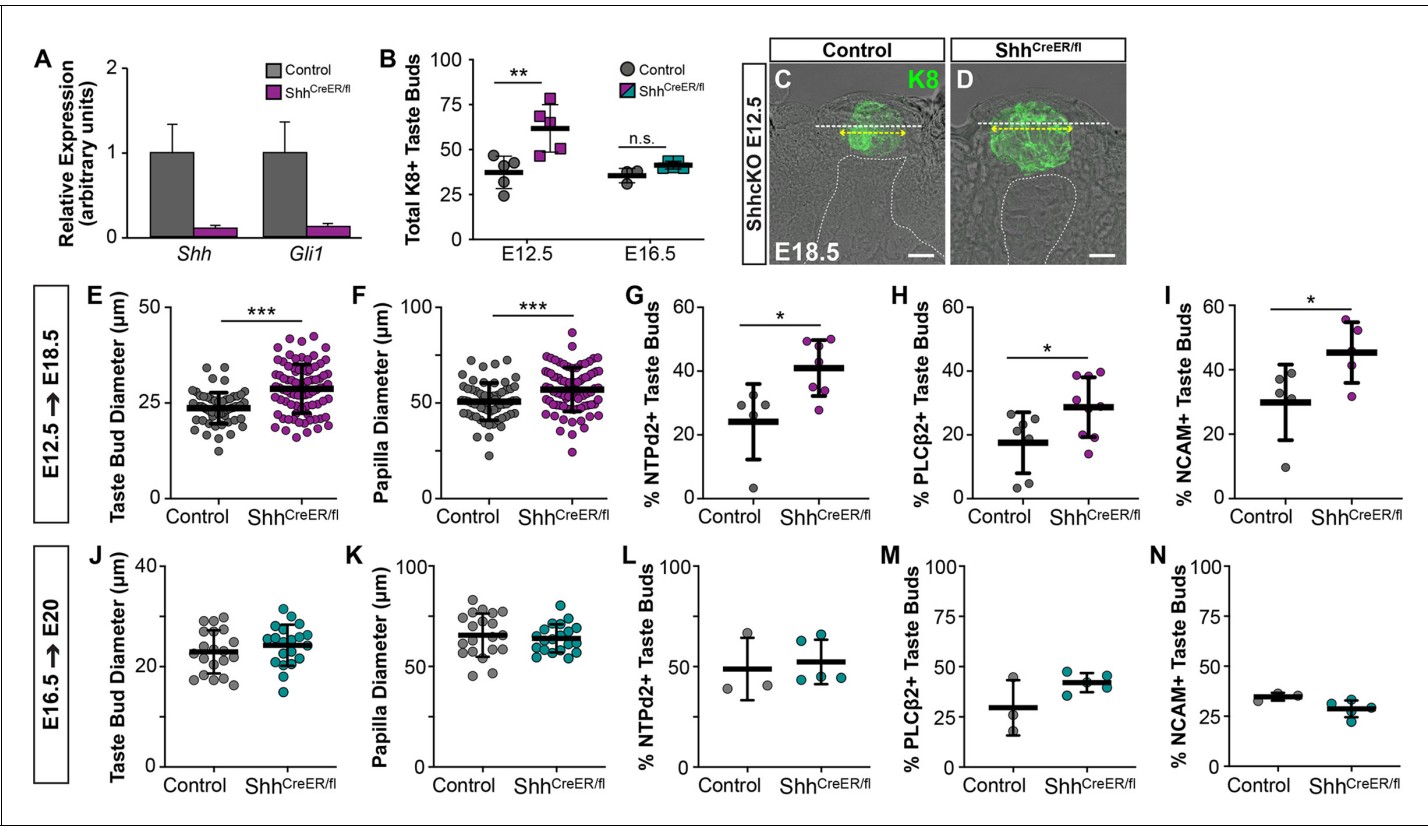

**Figure 3.** The timing of genetic deletion of Shh determines its impact on taste development in vivo. (A) *Shh* and *Gli1* are reduced in tongues of *Shh^CreERT2/fl* embryos at E18.5, induced at E13.5. (B) SHH deletion at E12.5 results in more KRT8+ taste buds at E18.5, but deletion at E16.5 does not alter taste bud number (N = 3–5 embryos per timepoint per genotype; Student's t-test **p<0.01). (C, D) Genetic deletion of SHH at E12.5 increases KRT8+ (green) taste bud profile diameter (yellow dash with arrows) and FFP diameter (white dash). Dotted line indicates basement membrane. Scale bar: 10 μm. (E, F) Taste bud and papilla diameter increases significantly in *Shh^CreERT2/fl* induced at E12.5 and assayed at E18.5 (N = 3–4 embryos per genotype, 20 taste buds per animal, Students t-test, ***p<0.001, **p<0.01). (G–I) The proportion of E18.5 taste buds housing type I (NTPdase2+), II (PLCß2+), and III (NCAM+) TRCs is significantly increased by SHH deletion at E12.5 (N = 5–8 animals per genotype, Student t-test, *p<0.05). (J, K) SHH deletion at E16.5 does not change taste bud number or size assayed at E20. (L–N) SHH deletion at E16.5 does not lead to precocious differentiation of type I (NTPdase2+), II (PLCß2+), and III (NCAM+) TRCs at E20. Black bars: mean ± SD for all plots.

The online version of this article includes the following figure supplement(s) for figure 3:

**Figure supplement 1.** Taste receptor cell differentiation begins before birth.

target transcriptional regulator, *Gli1* (*Figure 3*). In Shh-ShhcKO embyos induced at E12.5 and analyzed E18.5, the total number and size of KRT8+ taste buds and FFP were significantly increased (*Figure 3B–F*). Additionally, E12.5 Shh-ShhcKO significantly increased the proportion of taste buds containing differentiated type I (NTPdase2+), II (PLCß2+), and III (NCAM+) TRCs compared to *Shh^{wt/fl}* littermate controls (*Figure 3G–I*), indicating that Shh reduction during placode specification accelerated taste bud differentiation. This outcome is likely due to the loss of Shh's repressive role on Wnt/ß-catenin signaling during placode specification that in turn leads to precocious TRC differentiation (*Iwatsuki et al., 2007*; *Thirumangalathu and Barlow, 2015*).

Pharmacologic inhibition of the Shh pathway effector, Smoothened, in embryonic rat tongue explants suggests that Shh no longer regulates placode development once these structures have formed (*Liu et al., 2004*). Consistent with this, ShhcKO induced at E16.5 did not alter the number or size of FFP or KRT8+ taste buds (*Figure 3B,J,K*). Moreover, E16.5 induction of ShhcKO did not accelerate TRC differentiation; similar proportions of taste buds housed type I (NTPdase2+), II (PLCß2+), and III (NCAM+) TRCs in ShhcKO and control tongues at E20 (*Figure 3L-N*). Our in vivo results confirm and expand upon in vitro studies showing taste epithelium is not sensitive to Shh pathway inhibition in late gestation (*Liu et al., 2004*) (but see *Cohen et al., 2019*).

In contrast to its repressive function in embryos, in adult lingual epithelium misexpression of SHH induces ectopic taste buds in non-taste lingual epithelium (*Castillo et al., 2014*). Thus, we next asked if SHH promotes taste fate at birth, coinciding with the onset of taste cell production by KRT14+ progenitors. *Krt14^{CreERT2}*;*Rosa^{SHHcKI-IRES-nVenus}* (K14-SHHcKI) was induced in neonatal mice and tongues analyzed after 14 days. In addition to SHH, *Rosa^{SHHcKI-IRES-nVenus}* drives expression of nuclear Venus (YFP) in SHH+ cells. As in adults (*Castillo et al., 2014*), neonatal *Krt14^{CreERT2}* recombination resulted in patches of SHH-YFP+ cells in both FFP and non-taste epithelium. When K14-SHHcKI was induced at P1 and assessed at P15, small numbers of ectopic KRT8+ cells were detected in SHH-YFP+ non-taste lingual epithelium (*Figure 4A,B*). P1 induction of K14-SHHcKI largely resulted in individual ectopic KRT8+ cells (*Figure 4A*, yellow arrow). Significantly more KRT8+ cell clusters formed in K14-SHHcKI tongues induced at P14 and harvested at P28 (*Figure 3C,D*), and these ectopic buds co-expressed markers of type I (NTPdase2), II (PLCß2), and III TRCs (5HT; *Huang et al., 2005*; *Figure 4E–G*). Notably, forced SHH expression postnatally had no effect on endogenous FFP taste bud number (*Figure 4B,D*). By contrast, embryonic induction of K14-SHHcKI at E12.5 did not lead to the formation of ectopic buds at P14, but as expected repressed endogenous FFP taste bud development (*Figure 4—figure supplement 1*).

## Neurally supplied SHH is not required for taste progenitor activation

In adult tongue, SHH is expressed by gustatory sensory neurons of the VIIth cranial ganglion as well as post-mitotic taste precursor cells that differentiate all three functional TRCs. We have shown these cell populations together supply SHH to support continual TRC differentiation (*Castillo-Azofeifa et al., 2017*; *Miura et al., 2003*; but see *Lu et al., 2018* and discussed below). However, embryonic taste placodes are not innervated at specification, hence the only early source of lingual SHH is placodal (*Hall et al., 1999*). Gustatory neurons develop by E10.5 (*Cordes, 2001*) and gustatory neurites first penetrate the taste epithelium at E14.5 (*Krimm, 2007*; *Lopez and Krimm, 2006*). When gustatory neurons start to express SHH has not been determined. Thus, we next asked whether SHHs functional shift from a repressor of taste fate to a driver of TRC differentiation is correlated with the onset of gustatory neuron supply of SHH to embryonic taste primordia.

Lineage tracing with *Shh^{CreERT2}*;*R26T^{tdTomato}* leads to ample labeling of gustatory innervation in adult mice after 2 weeks (*Castillo-Azofeifa et al., 2017*). Thus, we induced Shh-tomato lineage tracing in embryos during placode specification (E12.5) and quantified the number of tomato+ nerve fibers within the core of each FFP at P14. In addition to ample taste bud cell labeling as expected, Shh lineage tracing revealed that ~1/3rd of taste buds were contacted by one or more tomato+ neurites at P14 (*Figure 5A,C,D*). Thus, some gustatory neurons already express SHH as early as E12.5. When lineage tracing was induced at birth (P0), well after nerve contacts are established, and similarly examined at P14, the proportion of FFP with tomato+ neurites increased significantly and taste buds were contacted by more tomato+ neurites (*Figure 5B–D*). Taken together, these results indicate that some gustatory neurons express SHH embryonically, well before the shift in SHH function, which occurs at birth.

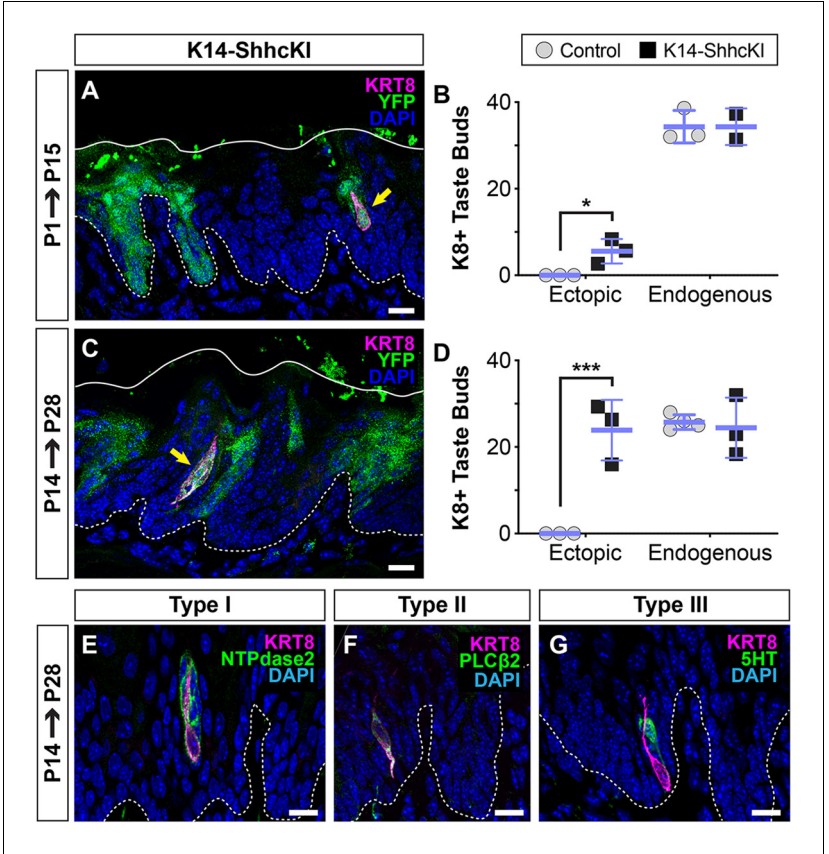

**Figure 4.** SHH promotes taste fate at birth. (**A, B**) KRT14-SHHcKI-YFP induction at P1 drives production of small numbers of ectopic KRT+ cells (magenta, yellow arrow) in patches of YFP+ non-taste lingual epithelium at P14 but does not impact the number of endogenous taste buds (n = 3 mice per timepoint) (Student's t-test *p<0.03). (**C, D**) KRT14-SHHcKI-YFP induction at P14 induces more and larger ectopic KRT8+ taste cell clusters at P28, with endogenous taste bud number unaffected (n = 3 mice per timepoint) (Student's t-test ***p<0.005). (**E–G**) Ectopic taste buds at P28 house cells expressing markers of type I (NTPdase2 **E**), II (PLCß2, **F**), and III (5HT, **G**) taste cells. (**A, C, E–G**) Dashed lines delimit the basement membrane; solid lines delimit the epithelial surface; yellow arrows indicate ectopic KRT8+ taste cells. Nuclei counter stained with DAPI. Scale bar: 10 μm.

The online version of this article includes the following figure supplement(s) for figure 4:

**Figure supplement 1.** Forced expression of SHH in embryonic lingual epithelium represses development of endogenous taste buds but does not induce ectopic taste buds.

---

We next tested if neuronal SHH is required for embryonic taste bud development and/or progenitor function postnatally. PHOX2b is a transcription factor expressed by developing cranial sensory neurons, including gustatory neurons that innervate FFP taste buds (*Ohman-Gault et al., 2017*; *Sajgo et al., 2016*). We used *Phox2b^{Cre}* (*Scott et al., 2011*) to constitutively delete SHH and drive tomato expression within PHOX2b+ neurons (*Phox2b^{Cre};Shh^{fl/fl};R26R^{tdTomato}*) (Phox2b-ShhcKO) (*Figure 5E,F*). When assessed at 10 weeks, taste buds in Phox2b-ShhcKO mice were indistinguishable from control *Phox2b^{Cre}; R26R^{tdTomato}* littermates; we found no differences in taste bud number, size (KRT8+ pixels/taste bud), or type II or III TRC differentiation (*Figure 5G–J*). These results suggest that neural SHH is dispensable for embryonic taste placode specification and postnatal taste bud differentiation, and are consistent with our previously findings in adults where epithelial SHH can compensate for loss of neuronal SHH to maintain adult taste buds (*Castillo-Azofeifa et al., 2017*).

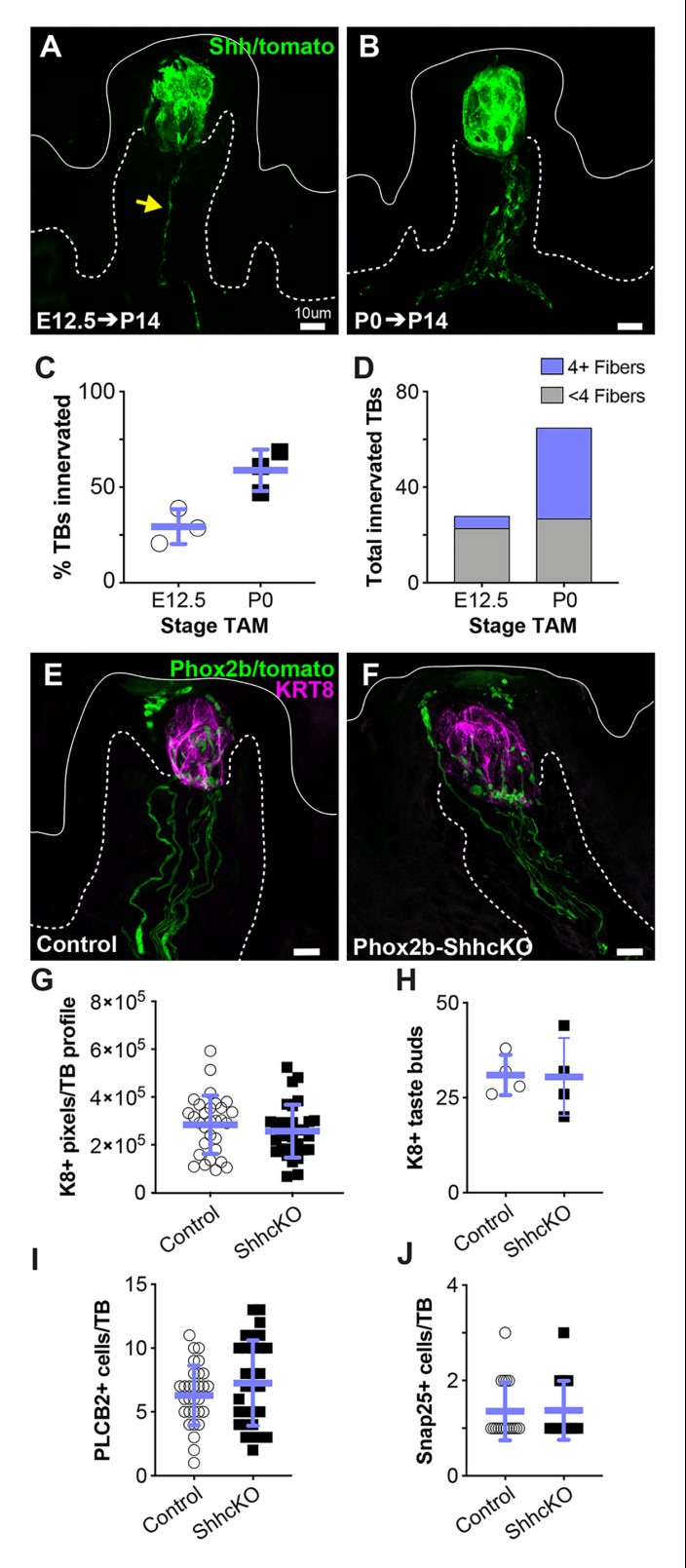

**Figure 5.** Embryonic gustatory neurons express SHH, but neural SHH is not required for taste development or postnatal taste bud maintenance. (**A**) $Shh^{CreERT2}$; $R26R^{tdTomato}$ (SHH-tomato) induction at E12.5 drives tomato expression (green) in taste cells and sparse gustatory neurites (yellow arrow) at P14. (**B**) Induction of SHH-tomato at P0 results in tomato+ taste cells and more numerous reporter-expressing gustatory neurites at P14. (**C**) More

*Figure 5 continued on next page*

*Figure 5 continued*

taste buds are innervated at P14 by P0 SHH-tomato lineage traced fibers compared to fibers lineage traced starting at E12.5. Student's *t*-test p=0.02. (**D**) At P14, individual taste buds are innervated more densely when lineage trace is initiated at P0 compared to initiation at E12.5 (n = 3 mice per stage, E12.5 induction – 98 taste buds, P0 induction – 111 taste buds). (**E, F**) *Phox2b^Cre* drives tomato expression (green) in gustatory neurites innervating KRT8+ taste buds (magenta) in control (*Phox2bCre;R26R^tdTomato*) and Phox2b-SHHcKO (*Phox2bCre; Shh^fl/fl;R26R^tdTomato*) mice. (**G, H**) Taste bud size (KRT8+ pixels per taste bud profile) and number do not differ between controls and Phox2b-SHHcKO mice at 10 weeks postnatal. (**I–K**) Phox2b-SHHcKO does not disrupt differentiation of PLCß2+ type II (**I**) or SNAP25+ type III (**J**) taste cells. Blue bars: mean ± SD (n = 3–4 mice per condition; 10 taste buds per animal; open and shaded shapes).

## Transcriptional profiling of lingual progenitors reveals additional candidates that function downstream of SHH in TRC renewal

The transcription factor SOX2 plays a critical role in taste bud development and renewal and is required downstream of SHH for adult taste bud differentiation (*Castillo-Azofeifa et al., 2018*; *Ohmoto et al., 2020*; *Okubo et al., 2006*). At P0, SOX2 is highly expressed in and around KRT8 + taste buds (*Figure 6A*, yellow arrows), moderately expressed by FFP basal progenitors (*Figure 6A*, yellow arrowheads), and dimly expressed by basal keratinocytes of the interpapilliary non-taste epithelium (*Figure 6A*, white arrowheads; see *Nakayama et al., 2015*), an expression pattern that is recapitulated in *SOX2-EGFP* (SOX2-GFP) reporter mice (*Figure 6B*; *Okubo et al., 2006*; *Taranova et al., 2006*). This expression pattern has led to the suggestion that high SOX2 is associated with taste fate, while moderate/dim SOX2 marks non-taste progenitors (*Castillo-Azofeifa et al., 2018*; *Nakayama et al., 2015*; *Okubo et al., 2006*). Thus, we sought to compare the transcriptional profiles of SOX2-GFP high vs low expressing epithelial cell populations at P0 to identify genetic pathways associated with activation of KRT14+ progenitors.

Dissociated SOX2-GFP lingual epithelial cells from 65 pups (10 litters) were FAC sorted into GFP^high and GFP^low populations (*Figure 6C,D*), and RNA isolated from ~100,000 cells per sorted cell population for gene profiling. Confirming the efficacy of cell sorting, *Sox2* was expressed 2× higher in GFP^high cells compared to GFP^low cells by qPCR (*Figure 6E*). To identify candidate genes regulating epithelial cell fate, we concentrated our RNAseq analyses on differentially expressed protein-coding genes (DEGs) enriched in GFP^high or GFP^low cells (*Supplementary files 1a–c*). Using a 5 FPKM threshold and enrichment of 1.5-fold or greater expression over the opposing population, we identified 1032 GFP^high and 970 GFP^low DEGs (*Figure 6—figure supplement 1*, *Supplementary files 1b, 1c*). As expected, taste and progenitor-associated genes were enriched in GFP^high cells, including *Slc6a11*, *Shh*, *Prox1*, *Lrmp*, *Ptch2*, *Slc6a13*, *Lgr5*, *Rarb*, *Six1*, *Bdnf*, *Wnt10b*, *Krt8*, *Krt19*, *Krt7*, and *Hes6* (*Supplementary file 1b*). The top 50 GFP^high DEGs also included genes broadly associated with epithelial development; *Gpa33*, *Stfa1*, *Krt75*, *Krt4*, and *Elf5* (*Figure 6—figure supplement 1*). By contrast, GFP^low DEGs included markers of both epithelium (*Lyg1*, *Krtap3-1*, *Krt84*, *Krt36*, *Krt83*, *Krtap13-1*, *Gfra2*, *Itga8*, *Krt81*) and connective tissue (*Foxd3*, *Sox10*, *Mpz*, *Ednrb*, *Zeb2*, *Cdh6*) (*Figure 6—figure supplement 1*, *Supplementary file 1c*). As some lingual mesenchymal cells appear to express low levels of SOX2 or *Sox2-GFP* (*Figure 6A,B* asterisks, and see *Arnold et al., 2011*), these cells were likely included in the profiled SOX2-GFP^low population. Thus, SOX2-GFP^high cells comprise an epithelial taste/progenitor enriched population, while SOX2-GFP^low cells represent a mixed population of lingual epithelium and connective tissue. These conclusions were further supported by Gene Ontology (GO) analyses (*Figure 6F*, *Supplementary files 1d, 1e*). GO terms for both populations included biological processes associated with developing epithelia (i.e. *Epithelium development*, *Cell migration*, *Cell adhesion*). SOX2-GFP^low analyses also included terms for non-epithelial tissues (*Vasculature development*, *Mesenchyme development*). While both datasets triggered the term *Sensory organ development*, only the GFP^high dataset was associated with the system-specific terms *Tongue development* and *Tongue morphogenesis*. In all, SOX2-GFP-based fluorescence-activated cell sorting (FACS) allowed for efficient isolation of differentiating taste and progenitor cells for gene expression profiling.

As expected, Hh pathway components were expressed by both GFP^high and GFP^low cells (*Figure 6G*). Consistent with previous reports, *Shh*, *Ptch1*, and *Gli1* were enriched in GFP^high compared to GFP^low cells (*Figure 6G,I*; *Hall et al., 1999*; *Liu et al., 2013*). Additionally, the

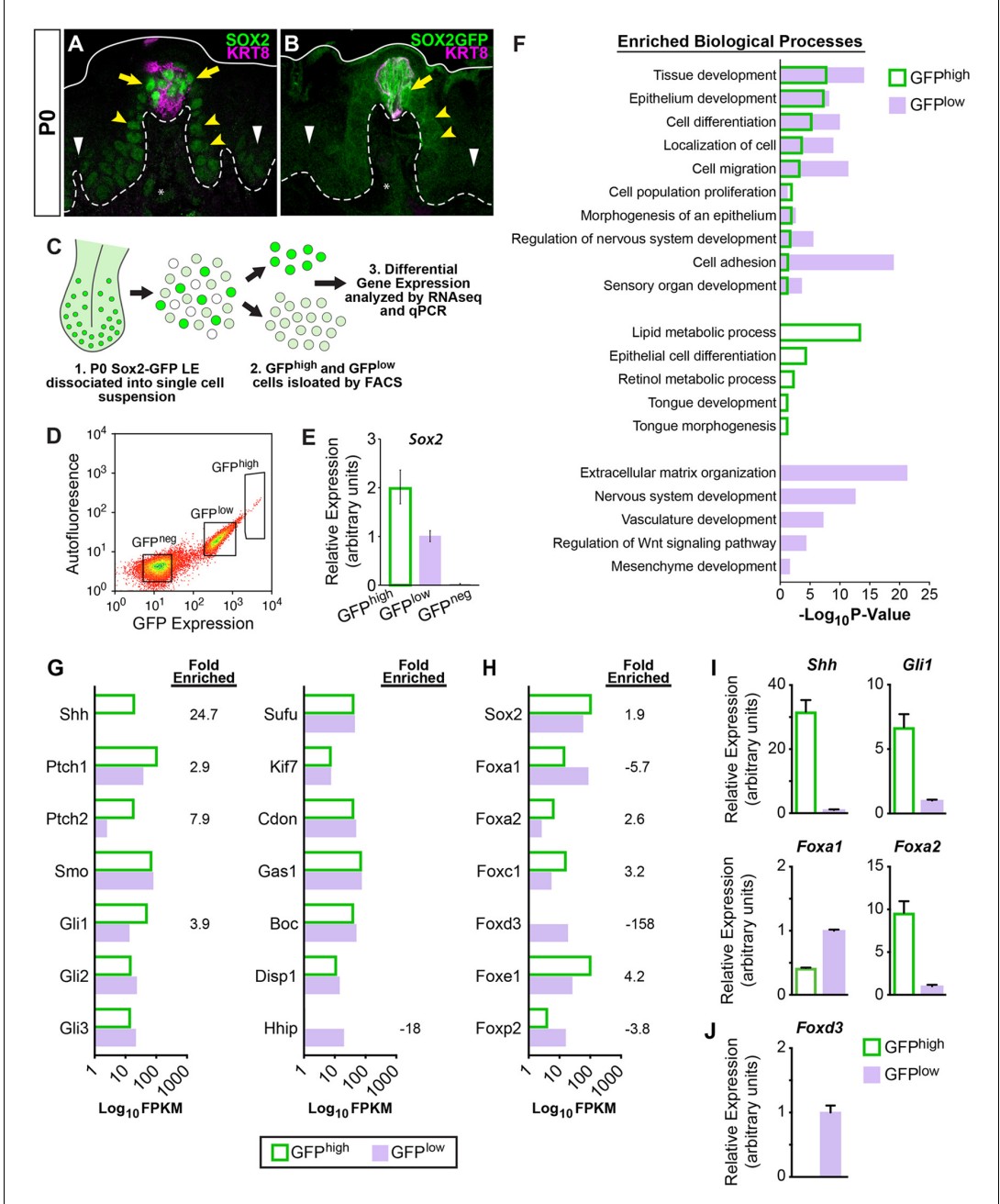

**Figure 6.** SHH pathway associated genes are differentially expressed in SOX2-GFP lingual epithelial cells. (**A, B**) At P0, SOX2 and SOX2-GFP (green) are expressed highly in KRT8+ taste buds (magenta) and perigemmal cells (yellow arrows), at lower levels in FFP epithelium (yellow arrowheads) and least intensely in non-taste epithelial basal cells (white arrowheads). Dashed lines delimit basement membrane, solid lines delimit apical epithelial surface. (**C**) Experimental procedure to isolate SOX2-GFP^high and SOX2-GFP^low lingual epithelial cells of P0 pups for RNAseq and qPCR analysis. (**D**) SOX2-GFP^high, SOX2-GFP^low, and SOX2-GFP^neg FAC sorted lingual epithelial cells were collected in discrete fluorescence bins from an expression continuum. (**E**) qPCR for *Sox2* confirms expression is highest in SOX2-GFP^high cells and absent in SOX2-GFP^neg cells. (**F**) GO term analysis of differentially expressed genes revealed processes enriched in SOX2-GFP^high vs SOX2-GFP^low populations (see text for details). (**G**) SHH pathway genes are enriched in SOX2-GFP epithelial cells. (**H**) Transcription factors associated with SHH signaling are differentially expressed in SOX2-GFP^high vs SOX2-GFP^low epithelial cells. (**I**) qPCR for SHH pathway associated genes confirms *Shh*, *Gli1*, and *Foxa2* are upregulated in SOX2-GFP^high cells, while *Foxa1* is more highly expressed in SOX2-GFP^low cells. (**J**) *Foxd3* is expressed only in the SOX2-GFP^low cells, consistent with the inclusion of mesenchymal cells in this population. The online version of this article includes the following figure supplement(s) for figure 6:

**Figure supplement 1.** Summary of SOX2-GFP^high vs SOX2-GFP^low differentially expressed genes.

*Figure 6 continued on next page*

Figure 6 continued

**Figure supplement 2.** Changes in expression of SOX2, FOXA2, and FOXA1 in developing FFP and taste buds may underlie changes in the lingual epithelial response to HH signaling.

transmembrane receptor *Ptch2* was highly enriched in the GFP^high^ population, while Hh-interacting protein *Hhip* was highly enriched in GFP^low^ cells. The role of these latter Hh pathway genes has not been explored in taste epithelium.

To identify specific candidate transcriptional regulators acting downstream of SHH within P0 lingual epithelium, we performed unbiased motif enrichment analysis on the GFP^high^ DEG dataset. Regions of interest (ROIs), which included 1000 kb upstream and 200 bp downstream of the transcription start site (TSS), were generated for each gene, and common regulatory sequences were evaluated in MEME-Suite AME (*McLeay and Bailey, 2010*). Of the 201 transcription factors identified by AME, only 123 were expressed within our P0 dataset (*Supplementary file 1f*). A literature search revealed ~20% of these candidates were associated directly or indirectly with the SHH pathway, including Sox2, FoxA1, and FoxA2.

In our dataset, *Sox2* and *Foxa2* were enriched within the GFP^high^ population (1.9× and 2.6×, respectively), while *Foxa1* was enriched in GFP^low^ cells (−5.65×) (*Figure 6H,I*). Notably, *Foxd3*, a canonical mesenchyme gene, was expressed exclusively in GFP^low^ cells (*Figure 6H,J*). From published reports, FOXA2 and FOXA1 are broadly expressed in developing lingual epithelium at E13.5–15.5, and FOXA1 maintains broad expression at E18.5; by contrast, at E18.5, FOXA2 expression is absent from the general tongue epithelium and restricted to subsets of cells within taste buds and to clusters of keratinocytes at the base of FFP (*Besnard et al., 2004*; *Luo et al., 2009*). To more precisely define the spatiotemporal expression of FOXA1 and FOXA2 in developing taste vs non-taste epithelium, we compared SOX2, FOXA2, and FOXA1 expression at E13.5 during placode specification when SHH signaling represses taste fate, at E16.5 when taste development is SHH insensitive, and at P0 as taste cell renewal begins and SHH promotes taste fate.

At E13.5, SOX2 is more highly expressed in KRT8+ taste placodes than by surrounding non-taste epithelium (*Nakayama et al., 2015*; *Okubo et al., 2006*; *Figure 6—figure supplement 2A*, asterisk and arrowheads). FOXA2 and FOXA1 are both expressed broadly by taste placode and non-taste epithelium (*Figure 6—figure supplement 2B,C*. placode: asterisk, non-taste epithelium: arrowheads). At E16.5, SOX2 expression remains robust in taste primordia and perigemmal cells adjacent to buds (*Figure 6—figure supplement 2D*, yellow arrows) but is lower in FFP and non-taste epithelial basal cells (white arrows and arrowheads). At E16.5, FOXA2 expression has altered, and is most intense in KRT8+ taste cells, moderate at the base of each FFP, and dim in non-taste epithelium (*Figure 6—figure supplement 2E*, white arrows and arrowheads); FOXA2 expression is absent from perigemmal cells (*Figure 6—figure supplement 2E*, yellow arrowheads). FOXA1 expression at E16.5 has also shifted and is evident at low levels in sparse taste cells, absent in perigemmal cells (*Figure 6—figure supplement 2F*, yellow arrowheads), but robust in the developing filiform papilla and interpapillary non-taste epithelium (fi, ip). At P0, the pattern of SOX2 and FOXA2 in taste buds and FFP was mostly unchanged from E16.5 (*Figure 6—figure supplement 2G,H*), although FOXA2 was significantly diminished in non-taste epithelium. FOXA1 expression remained low in subsets of taste cells and largely lacking in FFP keratinocytes, but strong in filiform papillae and interpapillary epithelium (*Figure 6—figure supplement 2I*). We conjecture that changes in expression of these three transcription factors could contribute to the transformation of the lingual epithelial response to SHH from an embryonic repressor to a taste fate promoter in adults.

Induction of K14-SHHcKI in adult lingual epithelium leads to formation of ectopic taste buds, including induction of high SOX2 within and moderate SOX2 expression around ectopic buds (*Castillo et al., 2014*). To determine whether changes in FoxA1 and FoxA2 expression also occur, we induced K14-SHHcKI at P1 and analyzed *Sox2*, *Foxa2*, and *Foxa1* expression at P15. Quantitative RT-PCR of *Shh* and *Gli1* revealed Hh signaling was highly elevated in K14-SHHcKI tongues compared to controls, as was *Sox2* (*Figure 7A*), as expected. Likewise, *Foxa2* was highly upregulated by K14-ShhcKI. However, *Foxa1* was significantly reduced by SHH overexpression (*Figure 7B*).

As expected K14-SHHcKI induction at P0 resulted in mosaic expression of SHH-YFP+ cell patches throughout an otherwise normal lingual epithelium at P15. In control regions of mutant tongues,

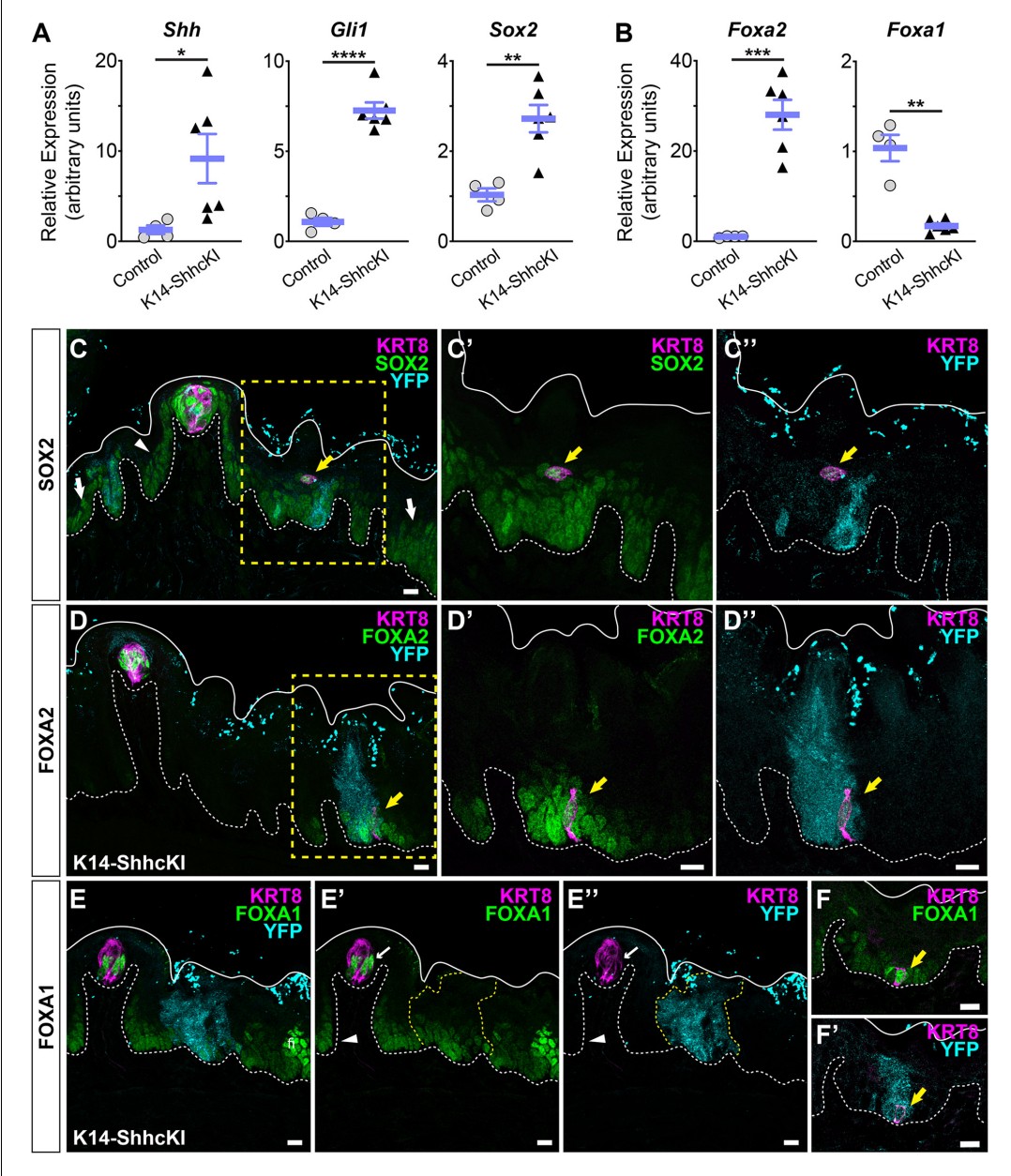

**Figure 7.** Sox2, FoxA2, and FoxA1 expression in lingual epithelium are altered by postnatal induction of SHH. (**A**) *Shh*, *Gli1*, and *Sox2* are increased in lingual epithelium harvested from P14 *Krt14^CreERT2^;ShhcKI-YFP* (KRT14-SHHcKI) pups induced with tamoxifen at P0. (**B**) *Foxa2* expression is increased in SHHcKI epithelium, while *Foxa1* is significantly reduced. (**A**, **B**): n = 4–6 mice per genotype, Student's t-test *p<0.05; **p<0.01; ***p<0.001; ****p<0.0001. (**C–C''**) SOX2 expression (green) is upregulated in and around patches of SHH-YFP+ cells (teal) in lingual epithelium of P14 mice induced with tamoxifen at P0. Ectopic KRT8+ taste cells (magenta) express elevated SOX2 (yellow arrow). (**D–D''**) FOXA2 is upregulated in and around patches of SHHcKI-YFP+ cells and is detected in ectopic KRT8+ taste cells (yellow arrow). (**E–E''**) FOXA1 expression appears unaffected in and around patches of SHHcKI-YFP+ cells (dashed yellow lines) but is also detected in occasional ectopic KRT8+ taste cells in SHHcKI-YFP+ domains (**F**, **F'**). Scale bars: 10 µm. Basement membrane delimited by dashed white line; solid white lines mark epithelial surface.

SOX2 expression at P15 resembles that reported in adult mice (*Figure 7C*; FFP basal keratinocytes, arrowhead), and non-taste basal keratinocytes (white arrows) (*Castillo-Azofeifa et al., 2018*; *Okubo et al., 2006*; *Suzuki, 2008*). In and around SHHcKI-YFP domains, SOX2 expression was detected in ectopic KRT8+ cells (yellow arrow) surrounded by dim SOX2+ cells (*Figure 7C',C''*), similar to the epithelial response observed in adult mice (*Castillo et al., 2014*).

When examined by immunofluorescence, K14-SHHcKI induction at P0 resulted in mosaic expression of SHH-YFP+ cell patches throughout an otherwise normal lingual epithelium at P15. Alterations in SOX2 expression within these YFP+ domains were similar to those previously reported in adult mice; elevated SOX2 expression was detected in ectopic KRT8+ cells (yellow arrow) surrounded by dim SOX2+ cells (*Figure 7C–C''*; *Castillo et al., 2014*). As in adult mice, SOX2 expression was unaltered in YFP[neg] control domains at P15 (*Castillo et al., 2014*).

Similarly, P0 induction of K14-SHHcKI at P0 led to ectopic FOXA2 expression at P15. Within YFP[neg] 'control' domains, FOXA2 was strongly expressed by KRT8+ taste buds and entirely absent from non-taste epithelium (*Figure 7D*). In contrast, FOXA2 expression was increased in and around induced SHH-YFP+ cell patches, including within ectopic KRT8+ taste cells (*Figure 7D–D''*, yellow arrow). Within YFP[neg] regions at P15, FOXA1 was expressed by small numbers of taste cells and by keratinocytes of filiform papillae and interpapillary epithelium (*Figure 7E–E'*). In contrast to SOX2 and FOXA2, a clear change in FOXA1 expression was not apparent in and around SHH-YFP+ cell patches (*Figure 7E',E''* yellow dash), but its expression was occasionally observed in ectopic KRT8 + taste cells (*Figure 7F,F'*). We attempted to quantify FOXA1 expression within and around SHH-GFP+ epithelium but were not able to discern any pattern (data not shown). Because the *Foxa1* qRT-PCR result was so robust (see *Figure 7B*), we surmise SHH regulation of FOXA1 is likely complex.

Given expression of these three transcription factors is affected by SHH, we next used motif analysis to identify potential target genes that may function downstream of SOX2, FOXA2, and/or FOXA1 in TRC differentiation. Within the SOX2-GFP[high] dataset, 45.6% of DEG ROIs contained a SOX2 binding motif, 38.4% a FOXA2 motif, and 24.8% a FOXA1 motif; 143 DEG ROIs contained motifs of all three transcription factors, while 90 DEGs had motifs for SOX2 and FOXA2, 102 DEGs had motifs for FOXA1 and FOXA2, and only 6 DEGs had motifs for SOX2 and FOXA1 (*Figure 8A Supplementary file 1g*). We next asked what biological processes might be regulated by these different combinations of transcription factors using GO analysis of the short list of DEGs with motifs for each combination of SOX2, FOXA2, and/or FOXA1 (*Supplementary file 1h*). No significant biological GO terms were associated with genes regulated by only FOXA2 or FOXA1, the combination of SOX2;FOXA1, or all three genes. Both SOX2 and SOX2;FOXA2 DEGs were mildly enriched for a small number of developmental and metabolic processes. However, FOXA2;FOXA1 DEGs were strongly enriched for terms including *Regulation of cell adhesion* and *Positive regulation of cell migration* (*Supplementary file 1i*). Among genes common to both GO terms were *Pdpn* (*Podoplanin*), *Plet1* (*Placenta expressed transcript 1*), *Cxcl10*, and *Tgfb2*, while *Runx1*, *Fbln2* (*Fibulin 2*), and *Ephb4* were unique to *Cell adhesion*. Our in silico results suggested that these genes may function in taste progenitor production of new taste cells at P0.

We next assessed if expression levels of these genes were altered in control versus SHHcKI tongues. We found expression of *cell adhesion* genes (*Runx1*, *Ephb4*, and *Fbln2*) were significantly increased in response to SHH (*Figure 8B*), as were genes common to both GO terms (*Plet1*, *Pdpn*, and *Cxcl10*) although only the increase in *Plet1* was significant (*Figure 8C*). By contrast, *Tgfb2* was significantly reduced by SHH (*Figure 8D*). These data support a model where SHH promotes TRC differentiation via regulation of FOXA1 and FOXA2, which in turn regulate genes involved in cell adhesion and cell movement.

## Discussion

Using inducible lineage tracing, we show that non-taste lingual epithelium and taste buds have a common embryonic origin from KRT14+ basal cells and that once placodes have formed, embryonic KRT14+ lingual progenitors do not contribute new cells to taste bud primordia; instead, generation of new TRCs from KRT14+ cells commences at birth and ramps up rapidly in the first two postnatal weeks. Additionally, we show that genetic induction of SHH expression represses taste bud development in embryonic tongue epithelium, while at birth, SHH promotes taste fate as progenitors are activated. This shift in SHH function does not correlate with the onset of SHH delivery from gustatory neurons, as these neurons express SHH before taste bud primordia are innervated, nor is neuronally supplied SHH required for postnatal taste bud development. Using RNAseq to profile SOX2-GFP[high] taste associated vs SOX2-GFP[low] expressing lingual epithelial cells, we identified known SHH target transcription factors FoxA2 and FoxA1 enriched in GFP[high] and GFP[low] populations, respectively. We

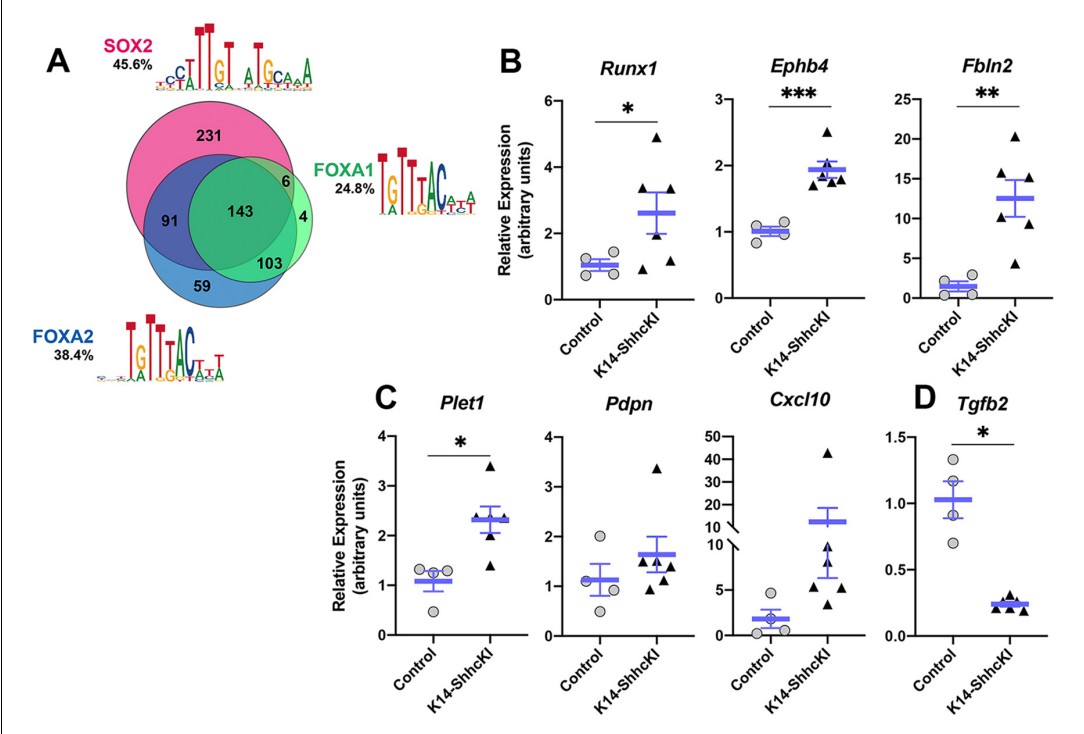

**Figure 8.** Expression of putative FOXA2;FOXA1 target genes associated with cell adhesion and movement is altered in response to forced SHH expression. (A) Motif analysis of DEGs in SOX2-GFP+ cells identified candidate target genes of SOX2, FOXA2, and FOXA1 in lingual epithelium at P0. See text for details. (B–D) Expression of putative target genes of FOXA2 and FOXA1 is altered in P14 lingual epithelium from KRT14-SHHcKI mice induced at P0. (B) *Runx1*, *Ephb4*, and *Fbln2* are significantly upregulated in KRT14-SHHcKI epithelium compared to controls. (C) *Plet1* is upregulated and *Pdpn* and *Cxcl10* trend upward following KRT14-SHHcKI induction. (D) *Tgfb2* is significantly downregulated by KRT14-SHHcKI. (B–D): n = 4–6 mice per genotype, Student's t-test, *p<0.05, **p<0.005, ***p<0.0005.

find FOXA2 and FOXA1, in addition to known expression of SOX2, are expressed in and around taste buds at P0 and their expression is altered in response to forced SHH expression. Additionally, we used motif analysis to identify potential targets of SOX2, FOXA1, and/or FOXA2. GO analysis of potential target genes indicates that FOXA1;FOXA2 together likely control a suite of genes that function in cell adhesion/migration. Finally, we show that expression of these genes is altered by forced SHH expression in vivo.

## Taste placodes and non-taste epithelium share a common KRT14 + progenitor at mid-gestation

In embryos, the surface ectoderm that gives rise to the epidermis is initially a single cell layer of KRT8/KRT18+ cells that by E9.5–10.5 also express KRT14/KRT5 and subsequently lose KRT8/18 expression (*Byrne et al., 1994*). Extensive lineage tracing studies of KRT14/5+ progenitors have firmly established that the epidermis and its specialized appendages, such as hair follicles and sebaceous glands, derive from this common embryonic population (*Pispa and Thesleff, 2003*; *Thesleff and Mikkola, 2014*). Similarly, we show in the developing tongue that KRT8 and KRT14 are broadly co-expressed at mid-gestation and confirm via KRT14 lineage tracing initiated at E12.5 that taste and non-taste epithelia share this common embryonic progenitor. A recent report using inducible Shh lineage tracing initiated at E11.0 identified an early Shh/KRT8 common progenitor population (*Kramer et al., 2019*), consistent with global epithelial expression of both SHH and KRT8 at this stage (*Echelard et al., 1993*; *Mbiene and Roberts, 2003*; *Zhu et al., 2014*). In contrast to our findings, however, KRT14-Cre lineage tracing presented in the same study resulted in reporter expression only in non-taste lingual epithelium, and not taste placodes or taste bud primordia as late as P1; lineage traced TRCs were only detected at P56 (*Kramer et al., 2019*) (also see *Bar et al., 2019*). It is likely that these different results reflect differences in the efficacy of the *Krt14Cre* alleles

employed, as randomly inserted transgenes like the different *Krt14* allele each study employed are notoriously prone to transgenerational silencing (*Haruyama et al., 2009*). Furthermore, gene promoter-specific transgene silencing can be incomplete, resulting in transgene expression that does not mirror the temporal expression of the endogenous gene product. Here, based on our demonstration of robust KRT14 protein expression at E12.5, we are confident that our lineage tracing at this stage with this *Krt14^CreERT2* allele (*Li et al., 2000*) is biologically accurate and allows us to conclude that taste placodes and non-taste epithelial cells share a common KRT14+ progenitor at midgestation.

## KRT14+ taste progenitor competency is activated at birth

Once specified, taste placodes/taste bud primordia do not increase in cell number for the remainder of embryogenesis. This stasis is consistent with numerous reports that taste placodes are mitotically quiescent, while the rest of the lingual epithelium actively proliferates (*Farbman and Mbiene, 1991*; *Liu et al., 2008*; *Mbiene and Roberts, 2003*; *Thirumangalathu and Barlow, 2015*). Rather we find taste cell number first increases in the first postnatal week, consistent with the demonstrated growth of postnatal taste buds by cell addition (*Hendricks et al., 2004*; *Krimm and Hill, 1998a*; *Krimm and Hill, 1998b*). Importantly, we show that these new TRCs arise from KRT14+ taste progenitors that are activated at birth and whose contribution steadily increases with postnatal age. Our data confirm and extend those of a previous report showing postnatal KRT14 lineage labeling of taste bud cells at P9 (*Okubo et al., 2009*). Although we can pinpoint when progenitors begin to generate new TRCs, it is not known when bona fide taste cell replacement begins. In rats, taste buds reach their mature size by P40 (*Hendricks et al., 2004*); thus, input and loss of TRCs must balance by this time. However, we showed previously that over half of SHH+ placode derived TRCs are lost by P42 in mice (*Thirumangalathu et al., 2009*). As taste buds continue to enlarge as embryonically derived TRCs are lost, taste bud cell replacement must begin before taste buds reach their mature size. That the rate of production of new TRCs is faster in the first 40 postnatal days than in adults may underpin the requirement for the growth of postnatal taste buds in the face of steady TRC loss (*Hendricks et al., 2004*).

## Embryonic SHH+ placodes and postnatal KRT14+ progenitors give rise to all TRC types

In adult mice, KRT14/5+ progenitors give rise to taste cells and non-taste epithelium (*Gaillard et al., 2015*; *Okubo et al., 2009*). Taste-fated daughter cells exit the cell cycle, enter taste buds, and express *Shh* (*Miura et al., 2006*). These SHH+ post-mitotic taste precursors then differentiate into each of the three major TRC types: I – glial-like support cells; II – sweet, bitter, or umami detectors; or III – sour sensors (*Miura et al., 2014*; *Takeda et al., 2013*). Here we show that KRT14+ progenitors also produce all three TRC types starting at birth. Previously, we found SHH+ placode cells differentiated into type I and II TRCs but did not detect type III cells among *Shh^CreERT2* lineage traced taste cells at postnatal stages (*Thirumangalathu et al., 2009*). Here using a tomato reporter allele, we demonstrate that SHH+ placodes, like adult SHH+ precursor cells, are fully competent to produce all TRC types. In addition, we show that embryonic SHH+ placodes and KRT14+ postnatal progenitors produce both type II and III TRCs assessed at early postnatal stages. Thus, we conjecture that both early embryonic and reactivated postnatal KRT14+ progenitors give rise to post-mitotic SHH+ cells that in turn differentiate into TRCs; the distinction in embryos is that SHH+ taste primordia remain poised for differentiation for several days until shortly before birth, while adult SHH+ precursors differentiate within 48–72 hr of their terminal division (*Miura et al., 2006*; *Miura et al., 2014*).

## SHH has opposing roles in embryonic vs adult lingual epithelium

SHH is a key regulator of taste epithelial development and renewal. In embryos, in vivo and in vitro data indicate that SHH represses taste fate during placode specification and patterning (*El Shahawy et al., 2017*; *Hall et al., 2003*; *Liu et al., 2004*; *Mistretta et al., 2003*). Here we confirm and expand upon these findings. In addition to more and larger taste papillae and taste buds, we find that early genetic deletion of SHH leads to precocious differentiation of TRCs. This is likely due to increased WNT signaling that is normally repressed by SHH to restrain placode size and number

(*Iwatsuki et al., 2007*; *Liu et al., 2007*). Additionally, excess WNT/ß-catenin signaling during placode specification drives precocious TRC differentiation, even in the absence of SHH (*Thirumangalathu and Barlow, 2015*), further suggesting that SHH deletion at mid-gestation leads to upregulated Wnt/ß-catenin signaling that in turn accelerates TRC differentiation. Once placodes are patterned, lingual epithelium is insensitive to Hh pathway inhibition in vitro (*Liu et al., 2004*), and we confirm this loss of sensitivity in vivo. Following genetic deletion of Shh at E16.5 after placode specification and patterning, taste bud and papilla size do not differ from controls, nor is TRC differentiation accelerated.

In adult tongue, SHH now promotes TRC differentiation. Genetic deletion or pharmacological inhibition of Hh pathway components in mice inhibits taste cell differentiation (*Castillo-Azofeifa et al., 2017*; *Castillo-Azofeifa et al., 2018*; *Kumari et al., 2015*; *Lu et al., 2018*), while forced SHH expression induces formation of ectopic taste buds comprising all three TRC types (*Castillo et al., 2014*). Here we show that the pro-taste function of SHH is evident at birth and coincides with the onset of KRT14+ progenitor contribution to taste buds, that is forced SHH expression at P0 causes formation of ectopic taste cells. By contrast, forced epithelial SHH expression does not induce TRC differentiation in embryonic lingual epithelium, but as expected, leads instead to smaller and fewer taste primordia, consistent with a recent report where repression of SHH expression in non-taste epithelium has been shown to be essential for taste bud maintenance in late gestation (*Bar et al., 2019*). Specifically, deletion of PRC-1 chromatin regulator genes in lingual epithelium leads to ectopic SHH expression that is associated with reduced FFP taste buds and, consistent with our study, does not induce ectopic taste buds.

We showed previously that TRC differentiation in adults is maintained by SHH supplied both by post-mitotic taste precursor cells within buds and by the gustatory innervation (*Castillo-Azofeifa et al., 2017*; *Lu et al., 2018*). Here we show that SHHs role in embryonic taste placode patterning is via its epithelial expression as taste bud primordia are only first innervated once placode specification and patterning are complete (*Lopez and Krimm, 2006*). Although we find that subsets of immature gustatory neurons taste express SHH before these fibers reach their taste placode targets (*Mbiene and Mistretta, 1997*; *Scott and Atkinson, 1998*), complete deletion of SHH in these neurons commencing as they first differentiate had no impact on taste placode specification or patterning in embryos, nor on taste cell differentiation in postnatal mice. In sum, these data suggest that as in adults, epithelial SHH can compensate for loss of a neural supply of SHH to support activation and maintenance of postnatal KRT14+ progenitor function (*Castillo-Azofeifa et al., 2017*). However, these and our previous results are at odds with recent study where using a presumed sensory neuron specific Cre driver (*Avil^Cre*), to delete neural SHH, led to loss of taste buds (*Lu et al., 2018*). Importantly, *Avil* is expressed in both type II and III TRCs of mice (*Sukumaran et al., 2017*), and *Avil^Cre* drives genetic reporter expression in both sensory neurons and taste buds (*Hasegawa et al., 2007*; *Zurborg et al., 2011*). Furthermore, our RNAseq data show that *Avil* is expressed in both low and high SOX2-GFP epithelial cells and is enriched in SOX2-GFP^high cells (*Supplementary file 1b*). Thus, *Avil^Cre* likely deletes SHH in both sensory neurons and taste bud cells – and therefore results from *Lu et al., 2018* are consistent with our model where both neural and epithelial SHH are required for TRC renewal.

## Transcriptional profiling implicates FoxA1 and FoxA2 in TRC renewal

The transcription factor SOX2 is required for TRC differentiation in neonates, and high levels of SOX2 have been suggested to be required for progenitor selection of taste over non-taste epithelial fate in adults (*Castillo-Azofeifa et al., 2018*; *Nakayama et al., 2015*; *Okubo et al., 2006*). RNAseq analysis of P0 SOX2-GFP^high and GFP^low lingual epithelial cells further support this hypothesis as the SOX2-GFP^high population is enriched with genes associated with TRC fate. Additionally, SOX2 functions downstream of SHH in adult taste renewal (*Castillo et al., 2014*; *Castillo-Azofeifa et al., 2018*), and at birth, many SHH pathway components are well expressed in SOX2-GFP^high cells. We identified a subset of transcription factors in our dataset linked with the SHH pathway, including FoxA2 and FoxA1, which are known to cooperate with and compensate for one another in development of numerous tissues, including brain and intestine (*Ang, 2009*; *Golson and Kaestner, 2016*; *Mavromatakis et al., 2011*; *Metzakopian et al., 2012*). As in other developing tissues, the relative expression of these two genes in developing lingual epithelium initially overlaps substantially and then partially segregates over time, suggesting that these changes could contribute to changes

in the lingual epithelial response to SHH. Consistent with a role in TRC renewal downstream of SHH, we found that ectopic TRCs induced by forced SHH expression also expressed FOXA2 and that forced epithelial SHH expression promoted FoxA2 and repressed FoxA1, thus identifying new candidates that likely coordinately function in TRC differentiation downstream of SHH.

To develop hypotheses as to the role these factors may play downstream of SHH in lingual epithelium, we identified potential gene targets via motif analysis for SOX2, FOXA2, and FOXA1 binding in our list of SOX2-GFP$^{high}$ DEGs. GO analysis revealed that FOXA1/2 may regulate a set of genes associated with cell migration and cell adhesion. All of these genes are enriched in SOX2-GFP$^{high}$ cells (*Supplementary file 1b*), and many are associated with cell adhesion changes and movement in both development and disease in a spectrum of tissues (*Nieto et al., 2016*). Thus, we hypothesize that TRC renewal may involve a partial EMT–MET process as has been proposed in developing and renewing hair follicles (*Hong et al., 2015*; *Jamora et al., 2005*; *Perez-Moreno et al., 2003*). Specifically, as KRT14+ progenitors activate, taste-fated daughter cells must modulate adhesion and motility to allow their exit from the non-taste epithelial adhesive environment to the specialized adhesive environment of taste buds thought to be critical for proper taste bud function (*Dando et al., 2015*; *Michlig et al., 2007*). A partial EMT–MET-like behavior has been observed for hair follicle cell renewal from bulge stem cells, where newly generated daughters integrate in distinct cell adhesive environments as they differentiate. The transcription factor OVOL2 is required for this process (*Hong et al., 2015*; *Lee et al., 2017*). Intriguingly, *Ovol2* is expressed at low levels in SOX2-GFP$^{high}$ and GFP$^{low}$ cells, while *Ovol1*, which regulates and may have overlapping function with Ovol2 in skin (*Teng et al., 2007*), is enriched 1.8-fold in P0 SOX2-GFP$^{high}$ cells. We also find that *Snai2*, a key EMT gene, is well expressed in both GFP$^{high}$ and GFP$^{low}$ populations. While expected in SOX2-GFP$^{low}$, which comprised epithelium and mesenchyme, *Snai2* expression in SOX2-GFP$^{high}$ cells is consistent with the idea that progenitors and taste-fated daughter cells activate EMT genes to enter taste buds (see *Wang et al., 2013*). Although we must further validate epithelial *Snai2* expression on tissue sections, expression of EMT genes by taste-relevant epithelial cells during TRC renewal may explain how lineage tracing using assumed 'mesenchymal' Cre driver alleles results in lineage labeling of subsets of TRCs in adult mice (*Boggs et al., 2016*; *Liu et al., 2012*).

## Materials and methods

### Animals

Male and female mice of mixed backgrounds were maintained, housed individually or in groups as approved, and processed at the University of Colorado Anschutz Medical Campus in accordance with approved protocols by the Institutional Animal Care and Use Committee. Mouse lines used include combinations of the following alleles or transgenes: Krt14$^{CreERT2}$ (*Li et al., 2000*), Shh$^{CreERT2}$ (Jax 005623), Phox2b$^{Cre}$ (Jax 016223), Shh$^{flox}$ (Jax 004293), R26R$^{Shh-IRES-nVenus}$ (*Castillo et al., 2014*), R26R$^{tdTomato}$ (Jax 007914), R26R$^{YFP}$(Jax 006148), and Sox2$^{GFP}$ (Jax 017592). Embryonic day (E) 0.5 was defined as midday on the day a mating plug was observed. Embryos were recovered on desired days of gestation and staged via eMouse Atlas Project Theiler criteria (http://www.emouseatlas.org/emap/ema/home.html).

### Embryonic and neonatal Cre induction

Timed-pregnant dams (intra-peritoneal) or neonatal pups (subcutaneous) received 100 mg/kg of tamoxifen dissolved in corn oil at indicated stages (T-5648, Sigma). Krt14$^{CreERT2}$;R26R$^{tdTomato}$, Krt14$^{CreERT2}$; R26R$^{YFP}$, Shh$^{CreERT2}$; R26R$^{tdTomato}$, and Shh$^{CreERT2/fl}$ mice were induced with a single dose of tamoxifen. Krt14$^{CreERT2}$; Rosa$^{SHHcKI-IRES-nVenus}$ animals and littermate controls received tamoxifen on two consecutive days, beginning at indicated stage.

### Immunostaining

Embryo heads (E12.0–E16.5) or peri- and postnatal tongues (E17.5–P28) were collected in ice-cold phosphate buffered saline (PBS) and fixed for 1 hr (light fix) or O/N (hard fix) at 4°C in 2% (vol/vol) paraformaldehyde (PFA) in PBS. Tissue was cryoprotected in 20% (wt/vol) sucrose at 4°C overnight and then submerged in OCT (Tissue-Tek, Sakura) and flash frozen on dry ice. Ten-week-old Phox2b-ShhcKO and control mice were processed using periodate–lysine–paraformaldehyde transcardial

perfusion as described in *Gaillard et al., 2015*. Serial 12 μm cryosections were collected on Super-Frost Plus slides (Fisher).

For immunostaining, slides were incubated in blocking solution (5% normal goat serum or donkey serum, 1% bovine serum albumin, 0.3% Triton X100 in 1× PBS, pH 7.3) for 1 hr at room temperature and then incubated with primary antisera in blocking solution overnight at 4°C (two overnights for transcription factors SOX2, FOXA1, and FOXA2). Sections were rinsed and incubated with secondary antisera in blocking solution for 1 hr at room temperature before nuclear counterstain with Draq5 (Abcam, ab108410) or DAPI (Invitrogen, D3571). Primary antibodies used were as follows: rat anti-KRT8 (1:250; Developmental Studies Hybridoma Bank, TROMA-I/AB_531826); rabbit anti-KRT14 (1:3500; Covance, PRB-155P); mouse IgG2a anti-E-Cadherin (1:100; BD Transduction Laboratories, 610181); chicken anti-GFP (1:500; Aves Labs, GFP-1020); rabbit anti-NTPDase2 (1:3000; CHUQ, mN2-36L); rabbit anti-PLCβ2 (1:1000; Santa Cruz, sc-206); rabbit anti-NCAM (1:1000; Millipore, AB5032); goat anti-CAR4 (1:1000; R and D Systems, AF2414); rabbit anti-5HT (1:1000; Immunostar, 20080); rabbit anti-SOX2 (1:100; Cell Signaling, 23064); rabbit anti-FOXA1 (1:100; Abcam, ab170933); and rabbit anti-FOXA2 (1:100; Cell Signaling, 3143). Alexa Fluor labeled secondary antibodies were used to visualize staining (1:1000; Thermo Fisher Scientific, A11008, A11010, A21245, A11006, A11081, A21247, A21208, A11039, A21137, A11055, A11056). To facilitate 5-HT detection, 5-hydroxy-L-tryptophan (Sigma-Aldrich H9772; 80 mg/kg, 6.4 mg/ml in 0.1 M PB, pH 7.4) was injected intraperitonially 1 hr prior to tissue collection. Slides were cover slipped with ProLong Gold mounting medium (Thermo Fisher).

E13.5 and 16.5 tissue collected for whole-mount analysis was fixed in 2% PFA for 2 hr at 4°C and then stored in PBS at 4°C until processing. Tongues were rinsed in room temperature PBS before undergoing methanol dehydration/rehydration (50% MeOH – 5 min; 70% MeOH – 5 min; 100% MeOH – 15 min; 70% MeOH – 5 min; 50% MeOH – 5 min; 25% MeOH – 5 min). Tissue was washed in 1× PBS with 0.2% Triton-X (PBST, 3 × 20 min washes) and then incubated in blocking solution for 2 hr at room temperature (see prior section for composition). Primary antisera incubation in blocking solution occurred for three overnights at 4°C with continuous agitation. On day 4, tissue was washed 3× 1 hr in PBST and then incubated in blocking solution containing secondary antisera and Draq5 nuclear counterstain for an additional two overnights at 4°C with continuous agitation. On day 6, tissue was washed 3× 1 hr in PBST and mounted on SuperFrost Plus slides (Fisher) with ProLong Gold mounting medium (Thermo Fisher).

## Thymidine analog birthdating

EdU (Life Technologies/Thermo Fisher) was reconstituted in PBS and administered at 50 μg/g of body weight to neonatal pups by a single subcutaneous injection. Click-iT Alexa Fluor 546 Kit (Life Technologies/Thermo Fisher) was used to detect incorporated EdU according to the manufacturer's specifications.

## Image acquisition and analysis

Anterior tongues were cut as 12 μm serial sections in six-slide sets, such that sections on each slide were separated by 72 μm. Approximately 16 sections were cut for each slide across the first ~1.2 mm of each tongue. After immunostaining, slides were de-identified and taste buds were mapped and quantified across each section/slide using a Zeiss Axioplan II microscope outfitted with a Retiga 4000R camera with Q-Capture Pro-7 software. Images of individual taste buds were acquired using a Leica TCS SP5 II laser scanning microscope with LASAF software and a Leica TCS SP8 laser-scanning confocal microscope with LAS X software. Images of sectioned taste buds were acquired via sequential confocal z-stacks with a z-step interval of 0.75 μm and 11–15 optical sections per z-stack. Immunolabeled cells were tallied by analyzing single optical sections and compressed z-stacks in LASAF, LAS X, and Image J software. All analyses were completed by investigators blinded to condition. KRT8 pixel quantification was conducted using an imstack toolbox developed in MATLAB as previously described (*Castillo-Azofeifa et al., 2017*). For taste bud innervation analysis, taste buds were classified as 'innervated' when at least one SHH-tomato+ nerve fiber was present within the FFP papillae core and it reached an apically situated tomato+/KRT8+ taste bud. When quantifying neurites innervating a given taste bud, all tomato+ neurites within the papilla core were counted.

For whole-mount E13.5 and E16.5 taste placode analysis, anterior tongues were imaged on a Leica TCS SP5 II laser scanning microscope with LASAF software. Taste placode maps were created from compressed sequential z-stacks consisting of 25 optical sections with a 0.5 µm z-step interval taken at 10x. Each KRT8+ taste placode within the first 1 mm of the dorsal surface of the anterior tongue was individually numbered, and 10 taste placodes/tongue were chosen at random (random. org) for quantification. For placode cell number and volume analysis, individual E13.5 and E16.5 taste placodes were imaged from their dorsal to ventral extent at 63× with a 1.5× digital zoom and a micro-sampling z-step interval of 0.25 µm. Placode z-stacks were analyzed in Imaris 8.4 (Oxford Instruments), using the ImarisCell tool. Briefly, a cube-shaped ROI was set in the x, y, and z planes using the KRT8 channel expression as a guide. E-cadherin expression was utilized by the software to delineate individual cell membranes within the ROI. KRT8 co-expression was then used to define intra-placodal versus extra-placodal cells. Each rendering was de-identified and manually reviewed to assure that cell boundaries were accurately delineated.

## RNA extraction and qRT-PCR

Freshly harvested tongues were rinsed and chilled in 1× dPBS. To isolate anterior tongue epithelium, enzyme peeling solution (1:1 mix of 2.5% pancreatin in L15 medium and 0.25% trypsin EDTA) was injected beneath the dorsal and ventral lingual epithelia and tongues incubated at room temperature for 45–60 min prior to manual dissection. Postnatal enzyme peeling solution contained 5 mg/ml Dispase II + 3 mg/ml Collagenase II in 1× dPBS, and tongues were incubated for 31 min at room temperature prior to manual dissection. Peeled tissue was snap frozen on dry ice and stored at −80° C.

Total RNA was extracted with a RNeasy Plus Micro kit (Qiagen, 74034) according to the manufacturer's instructions, and RNA quantity was measured by Nanodrop (ThermoFisher Scientific). mRNA was reverse transcribed using the iScript cDNA synthesis kit (Bio-Rad). SYBR Green-based qRT-PCR was performed by using Power SYBR Green PCR Master Mix reagent (Applied Biosystems) and gene-specific primers. qRT-PCRs were carried out in triplicate using a StepOne Plus Real-Time PCR System (Applied Biosystems, Life Technologies). Relative gene expression was analyzed using the ∆∆CT method (*Livak and Schmittgen, 2001*). The ribosomal gene *Rpl19* was used as an endogenous reference gene.

| Gene | Forward | Reverse | Accession numbers |
|---|---|---|---|
| Rpl19 | GGT CTG GTT GGA TCC CAA TG | CCC GGG AAT GGA CAG TCA | NM_009078.2 |
| Shh | AAG TAC GGC ATG CTG GCT CGC | GCC ACG GAG TTC TCT GCT TTC ACA G | NM_009170.3 |
| Gli1 | GGA AGT CCT ATT CAC GCC TTG A | CAA CCT TCT TGC TCA CAC ATG TAA G | NM_010296.2 |
| Sox2 | CCA GCG CAT GGA CAG CTA | GCT GCT CCT GCA TCA TGC T | NM_011443 |
| Foxa1 | GTG GAT CAT GGA CCT CTT CC | GCC ACC TTG ACG AAA CAA TC | NM_008259.3 |
| Foxa2 | AGC AGA GCC CCA ACA AGA TG | TCT GCC GGT AGA AAG GAA GGA | NM_001291065.1 |
| Foxd3 | CAG CAA CCG TTT TCC GTA CT | GGG TCC AGG GTC AGG TAG TT | NM_010425.3 |
| Foxe1 | AAC CTC ACC CTC AAC GAC TG | GCT TTC GAA CAT GTC CTC GG | NM_183298.1 |
| Cxcl10 | CTT CTG AAA GGT GAC CAG CC | GTC GCA CCT CCA CAT AGC TT | NM_021274.2 |
| Ephb4 | TAT GCC ACG ATA CGC TTC ACC | AGC TTC GCT CTC GTA ATA GAA GA | NM_001159571.1 |
| Fbln2 | AGT GGC CGT AAG TAT GCT GC | GGA AGC TGG TAG CAA ATG AGC | NM_007992.2 |
| Pdpn | ACC GTG CCA GTG TTG TTC TG | AGC ACC TGT GGT TGT TAT TTT GT | NM_001290822.1 |
| Plet1 | CAC TAT GGC TAA CGT CTC TGG | CTG TCG TCC TCC TTC ACT G | NM_029639.2 |
| Runx1 | GCA GGC AAC GAT GAA AAC TAC T | GCA ACT GTT GGC GGA TTT GTA | NM_001111021.2 |
| Snai2 | TGG TCA AGA AAC ATT CA ACG CC | GGT GAG GAT CTC TGG TTT TGG TA | NM_011415.2 |
| Tgfb2 | CTT CGA CGT GAC AGA CGC T | GCA GGG GCA GTG TAA ACT TAT T | NM_009367.4 |

## FACS and RNA extraction for RNAseq

Lingual epithelium was isolated from SOX2-GFP P0 mice using embryonic enzyme peeling solution and manual dissection as described above (45–60 min room temperature incubation in 1:1 mix of 2.5% pancreatin and 0.25% trypsin EDTA). Peeled tissue pieces were chopped with a sharp blade and then further digested in enzyme solution of 1 mg/ml Collagenase II and 1 mg/ml elastase in $1\times$ dPBS for 1 hr at 37°C. Digested tissue was manually triturated into single cells using a sterile glass pipette and then pelleted in DMEM/10% FBS to inhibit further enzyme digestion. Pellets were resuspended in FACS buffer (1 mM EDTA, 25 mM HEPES pH 7.0, 1% FBS, and $1\times$ $Ca^{2+}/Mg^{2+}$ free dPBS) and passed through a 30 µM nylon mesh filter to isolate single cells from remaining aggregates. FACS-based purification of SOX2-GFP cells was carried out on a MoFlo XDP70 (Gates Center Flow Cytometry Core, University of Colorado Anschutz Medical Campus) according to the green fluorescent protein signal (excitation 488 nm; emission 530 nm). Red fluorescent protein channel (582 nm) was used to gate out autofluorescent cells, and DAPI (450 nm) was used to gate out dead cells. Total RNA was extracted from FACS-isolated cell pools (GFP$^{high}$, GFP$^{low}$, GFP$^{neg}$) using the Arcturus Pico-Pure RNA Isolation Kit (Thermo Fisher #KIT0204) according to manufacturer's instructions.

## RNAseq analysis

Illumina HiSEQ libraries were prepared and sequenced by the Genomics and Microarray Core Facility at the University of Colorado Anschutz Medical Campus (Illumina HighSEQ4000, HT Mode, 1 × 50). Base calling and quality scoring were performed using Illumina Real-Time Analysis (RTA) v2.7.3 software. Files were demultiplexed and converted from BCL to FASTQ format using bcl2fastq2 v2.16.0.10 conversion software. Single-end reads were trimmed using Trimmomatic v0.36 and subsequently mapped to the mm10 mouse genome using GMAP/GSNAP v20141217. FPKM values were then obtained using Cufflinks v2.2.1. Analysis was conducted using R 3.5.3. Non-protein-coding genes and genes with no expression across all samples were removed prior to analysis. Genes with an absolute fold change greater than or equal to 1.5 and FPKM values greater than or equal to 5 in each sample were designated as differentially expressed (DEG).

## GO analysis

GO analysis of DEGs was performed in g:Profiler using user default settings (*Supplementary files 1d and 1e*), and GO enrichment analysis was powered by PANTHER (geneontology.org) (*Supplementary files 1h and i*).

## Motif enrichment analysis

ROIs that included 1 kb upstream and 200 bp downstream of the TSS were extracted for the 1032 differentially expressed SOX2-GFP$^{high}$ genes using the 'biomaRt' package for R. For genes with multiple isoforms, the APPRIS principal isoform was selected. In the few cases where multiple principal isoforms were annotated with APPRIS, the isoform with the TSS most upstream of the gene start site was used, resulting in one ROI for each differentially expressed gene. ROIs were converted to BED format and then FASTA format using Bedtools (version v2.26.0–129-gc8b58bc) and the Ensembl GRCm38 mouse genome (release 96).

FASTA files of ROIs were used as the input for AME (MEME-Suite, v5.0.5) and were tested against the HOCOMOCO mouse V11 motif database using default parameters but for the `'-control -shuffle-'` flag. AME outputs were parsed with a custom R script (*Larson, 2021*) to compare motif enrichment and gene expression data. Data are presented as the ratio of motif hits in the primary sequence divided by motif hits in the background (shuffled primary sequences).

## Statistical analyses

Normally distributed data were analyzed in Graphpad Prism eight software using parametric two-tailed Student's t-tests. Significance was taken as $p < 0.05$ with a confidence interval of 95%. Data are presented as mean ± standard deviation (SD) unless otherwise indicated.

## Acknowledgements

The authors thankthe Gates Center for Regenerative Medicine Flow Cytometry Core and the University of Colorado Cancer Center Genomics and Microarray Shared Resource supported by NCI P30CA046934 as well as Kelly Zaccone and Ernesto Salcedo for technical support. Thank you to Fred De Sauvage, Genentech, for providing the initial *ShhcKI-YFP* mice. Thank you also to our colleagues in the Rocky Mountain Taste and Smell Center as well as the Department of Craniofacial Biology at the University of Colorado Anschutz Medical Campus for constructive conversations and feedback throughout the duration of the study. This work was supported by grants from the National Institutes of Health/National Institute for Deafness and Other Communication Disorders to EJG (F32 DC015958) and to LAB (R01DC012383).

## Additional information

### Funding

| Funder | Grant reference number | Author |
|---|---|---|
| National Institute on Deafness and Other Communication Disorders | R01DC012383 | Linda A Barlow |
| National Institute on Deafness and Other Communication Disorders | F32DC015958 | Erin J Golden |

The funders had no role in study design, data collection and interpretation, or the decision to submit the work for publication.

### Author contributions

Erin J Golden, Conceptualization, Data curation, Formal analysis, Funding acquisition, Validation, Investigation, Methodology, Writing - original draft, Project administration; Eric D Larson, Data curation, Software, Formal analysis, Methodology, Writing - review and editing; Lauren A Shechtman, Data curation, Formal analysis, Visualization, Methodology, Writing - review and editing; G Devon Trahan, Data curation, Formal analysis, Writing - review and editing; Dany Gaillard, Data curation, Methodology, Writing - review and editing; Timothy J Fellin, Formal analysis, Methodology, Writing - review and editing; Jennifer K Scott, Data curation, Investigation, Methodology, Writing - review and editing; Kenneth L Jones, Data curation, Formal analysis, Supervision, Writing - review and editing; Linda A Barlow, Conceptualization, Data curation, Formal analysis, Supervision, Funding acquisition, Writing - original draft, Project administration, Writing - review and editing

### Author ORCIDs

Eric D Larson https://orcid.org/0000-0002-2881-4861
Dany Gaillard https://orcid.org/0000-0002-6875-025X
Linda A Barlow https://orcid.org/0000-0001-7998-2219

### Ethics

Animal experimentation: Male and female mice were maintained, bred, and embryos and pups at the University of Colorado Anschutz Medical Campus (CU AMC) in accordance with approved protocols #00150 and #52815(02)1C by the Institutional Animal Care and Use Committee at CU AMC. All animals were euthanized via chilling and $CO_2$ prior to tissue harvest to minimize suffering.

### Decision letter and Author response

Decision letter https://doi.org/10.7554/eLife.64013.sa1
Author response https://doi.org/10.7554/eLife.64013.sa2

## Additional files

### Supplementary files

• Supplementary file 1. Results of bioinformatic analyses of SOX2-GFP+ lingual epithelial cells from P0 mouse pups. (a) List of all differentially expressed protein-coding genes (DEGs) in SOX2-GFPlow and SOX2-GFPhigh lingual epithelial cells of neonatal mice at P0. Non-protein-coding genes and genes with no expression across all samples were removed prior to analysis. Genes with an absolute fold change $\geq$ 1.5 and FPKM $\geq$ 5 in each sample were designated as differentially expressed. (b) List of DEGs enriched in SOX2-GFP[high] lingual epithelial cells at P0. (c) List of DEGs enriched in SOX2-GFP[low] lingual epithelial cells at P0. (d) List of Gene Ontology (GO) biological process terms associated with genes enriched in SOX2-GFP[high] lingual epithelial cells of neonatal mice at P0. (e) List of GO biological process terms associated with genes enriched in SOX2-GFP[low] lingual epithelial cells at P0. (f) List of transcription factors with binding motifs enriched in SOX2-GFP[high] cells, including Sox2, Foxa2, and Foxa1.(g) Gene lists of potential SOX2 and/or FOXA2 and/or FOXA1 target DEGs enriched in SOX2-GFP[high] lingual epithelial cells at P0. (h) List of GO biological process terms associated with FOXA2 and/or FOXA1 and/or SOX2 target genes enriched in SOX2-GFP[lhigh] lingual epithelial cells. (i) Lists of SOX2-GFP[high] DEGs found in biological process GO terms '...cell adhesion' and '...cell motility' potentially regulated by FOXA1 and FOXA2.

• Transparent reporting form

### Data availability

Sequencing data have been deposited in GEO under accession code GSE159941 https://www.ncbi.nlm.nih.gov/geo/query/acc.cgi?acc=GSE159941.

The following dataset was generated:

| Author(s) | Year | Dataset title | Dataset URL | Database and Identifier |
|-----------|------|---------------|-------------|-------------------------|
| Golden EJ, Larson ED, Barlow LA | 2021 | Onset of taste bud cell renewal starts at birth and coincides with a shift in SHH function | https://www.ncbi.nlm.nih.gov/geo/query/acc.cgi?acc=GSE159941 | NCBI Gene Expression Omnibus, GSE159941 |

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
