## [Decision Letter]

**Acceptance summary:**

This convincing and well-written manuscript provides a further validation of previous findings of opposing role of Shh in taste cell development and renewal, while also adding new insights into downstream targets of Shh signaling. These data represent a very first step in understanding how Shh signaling regulates taste cell renewal mechanistically and provides a foundation for future exploration of Shh signaling.

**Decision letter after peer review:**

Thank you for submitting your article "Onset of taste bud cell renewal starts at birth and coincides with a shift in SHH function" for consideration by *eLife*. Your article has been reviewed by 2 peer reviewers, and the evaluation has been overseen by a Reviewing Editor and Kathryn Cheah as the Senior Editor. The following individual involved in review of your submission has agreed to reveal their identity: Lindsey MacPherson (Reviewer #2).

The reviewers have discussed the reviews with one another and the Reviewing Editor has drafted this decision to help you prepare a revised submission.

Summary:

This convincing and well-written manuscript provides a further validation of previous findings of opposing role of Shh in taste cell development and renewal, while also adding new insights into downstream targets of Shh signaling. These data represent a very first step in understanding how Shh signaling regulates taste cell renewal mechanistically, and provides a foundation for future exploration of Shh signaling.

Essential revisions:

1. Confirm if Shh is truly deleted in this model using qPCR, immunostaining, or in situ. Without that evidence, it remains possible that this model may be not adequate for deleting Shh.

2. To extend the current work in an effort to extend the impact of this work on the field, the reviewers thought this manuscript could address a potential conflict in the field. Recent work from the Barlow lab has shown that Sonic hedgehog from both nerves and epithelium is a key trophic factor for taste bud maintenance (https://dev.biologists.org/content/144/17/3054), and the Beachy lab showed that neuronal delivery of Hedgehog directs spatial patterning of taste organ regeneration (https://www.pnas.org/content/115/2/E200). Can the authors use their new mouse model to determine opportunity what happens in when Shh is ablated in all Shh-expressing cells using ShhCreERT2/fl mice to either support their previous conclusion and/or Beachy lab's conclusion?

3. The sentence starting on 341 states that "FOXA1 expression appears diminished". There needs to be quantification here. The authors could provide pixel intensity differences between the SHH-YFP+ patches and the surrounding epithelium to confirm diminished expression.

4. Discussion points to add to the text.

A. Over-expression of Shh (YFP) in Krt14 cells leads to generation of ectopic taste cells, as the authors reported previously. Yet, ectopic taste cells occur only infrequently in YFP+ patches (Figure 4). It would be interesting to discuss that. Are these ectopic taste cells are generated from Krt14-Shh cells that are ectopic taste progenitor cells? Alternatively, over-expression of Shh may stochastically convert lingual stem/progenitor cells to taste progenitor cells. These questions may be difficult to address. Nevertheless, the authors may want to discuss this intriguing observation.

5. Additional textual changes.

A. To place the problem in context for a broad readership, a schematic explaining the relationship between K14 and SHH in the developing taste bud; what is known and what your experimental data added. As written, the ideas are not well explained as a whole concept.

---

## [Author Response]

Essential revisions:1. Confirm if Shh is truly deleted in this model using qPCR, immunostaining, or in situ. Without that evidence, it remains possible that this model may be not adequate for deleting Shh.2. To extend the current work in an effort to extend the impact of this work on the field, the reviewers thought this manuscript could address a potential conflict in the field. Recent work from the Barlow lab has shown that Sonic hedgehog from both nerves and epithelium is a key trophic factor for taste bud maintenance (https://dev.biologists.org/content/144/17/3054), and the Beachy lab showed that neuronal delivery of Hedgehog directs spatial patterning of taste organ regeneration (https://www.pnas.org/content/115/2/E200). Can the authors use their new mouse model to determine opportunity what happens in when Shh is ablated in all Shh-expressing cells using ShhCreERT2/fl mice to either support their previous conclusion and/or Beachy lab's conclusion?

These first 2 points focus on figure 4 where we show deletion of Shh in gustatory neurons via *Phox2b^Cre^;Shh^fl/fl^* which is initiated embryonically in gustatory neurons before axons even enter the developing tongue, has no impact on taste bud formation or taste receptor cell differentiation in adult mice assessed at 10 weeks postnatal

We have spent many months trying to answer this critique with additional experiments.

Point 1. We no longer had the *Phox2b^Cre^;Shh^fl/fl^
*line in our colony, as we had to kill down most of our mouse lines in March 2020 when our university shut down. We received our review in October and quickly realized reconstituting that line would take months such that acquiring new samples for qPCR analysis was not an option. Instead, we attempted both conventional in situ and HCR for *Shh* on frozen sections of fixed control and mutant VII^th^ cranial nerve ganglia collected previously. Unfortunately, *Shh* mRNA was not detectable in controls using either method. This is consistent with transcriptome data of ganglion VII neurons (see BioRxiv https://doi.org/10.1101/812578 ), analyzed by one of us (Eric Larson), where *Shh* is very lowly expressed in subsets of gVII neurons. However, we still favor the view that Phox2b^Cre^ is an efficient allele that results in loss of gustatory neuron Shh, given its widespread use by others to efficiently delete specific genes in sensory neurons, including gVII (Donnelly et al., 2018; Tang et al., 2020).

A crucial point is that the Beachy lab used an *Avil^Cre^* allele (Hasegawa et al., 2007) assuming sensory neuron specificity of Shh deletion. However, both *Avil^Cre^* and *Avil^CreE^*^R^ alleles show reporter expression in mouse taste buds (Hasegawa et al., 2007; Zurborg et al., 2011). Further, our own RNAseq data show *Avil* is expressed in both low and high SOX2-GFP epithelial cells and is enriched in SOX2-GFP^high^ epithelial cells (Suppl Table 2); and *Avil* is expressed in mouse type II and III TRCs (Sukumaran et al., 2017). These finding support the interpretation that *Avil^Cre^* deletes Shh in both sensory neurons and taste bud cells – consistent with our conclusions in (Castillo-Azofeifa et al., 2017) where SHH in both tissue compartments supports TRC renewal.

Point 2. In (Castillo-Azofeifa et al., 2017), we employed the *Shh^CreER/fl^* model in an attempt to delete Shh in both taste buds and sensory neurons (previous Figure 5), treating mice with tamoxifen for 4 days and assessing taste buds at 35 days. In these experiments we detected no impact on taste buds despite lineage tracing in both ganglion cells and some cells in taste buds. However, the long chase after initial CreER induction likely allowed for production of new SHH^+^ taste bud cells from Shh^neg^ progenitors. In an earlier study (Miura et al., 2014), we explored a variety of dosing protocols to lineage label SHH^+^ taste bud cells, but were never able to fully activate CreER in all of these cells, even after treating mice for 21 days with tamoxifen in the drinking water. In fact, with tamoxifen-laced drinking water we observed reduced labeling in some taste fields, suggesting prolonged tamoxifen dosing reduced efficiency of this genetic model. Thus, using *Shh^CreER/fl^* as suggested by the reviewers is not a tractable approach to fully delete SHH in both epithelium and neurons. In fact, we arrived at this conclusion previously, instead we combined stereotaxic AAVCre brain injections with conditional genetic deletion in taste progenitors to delete Shh in gustatory neurons and epithelia in concert; this resulted in a complete loss of buds (Castillo-Azofeifa et al., 2017).

A synopsis of this argument is presented in Lines: 465-483.

3. The sentence starting on 341 states that "FOXA1 expression appears diminished". There needs to be quantification here. The authors could provide pixel intensity differences between the SHH-YFP+ patches and the surrounding epithelium to confirm diminished expression.

This proved difficult given regional variability in the extent and size of Shh-YFP+ patches. In ventral tongue FOXA1 expression appeared little affected by SHH-cKI, while in some areas in dorsal tongue it seemed that FOXA1 was potentially downregulated but may vary with the size of SHH-YFP+ patches; and if these occurred adjacent to and included fungiform epithelium or were located in non-taste epithelium. The pattern was simply too variable and complex to allow us to make any conclusions given the variability in the tongue samples we examined. Thus, we have softened our language in the text to limit our interpretation to *Foxa1* downregulation and leave open the extent of FOXA1 reduction as explained. (Lines: 330-346)

4. Discussion points to add to the text.A. Over-expression of Shh (YFP) in Krt14 cells leads to generation of ectopic taste cells, as the authors reported previously. Yet, ectopic taste cells occur only infrequently in YFP+ patches (Figure 4). It would be interesting to discuss that. Are these ectopic taste cells are generated from Krt14-Shh cells that are ectopic taste progenitor cells? Alternatively, over-expression of Shh may stochastically convert lingual stem/progenitor cells to taste progenitor cells. These questions may be difficult to address. Nevertheless, the authors may want to discuss this intriguing observation.

We know the lingual epithelium outside of taste buds is generated by a heterogeneous set of progenitors, but the extent to which one or more of these is taste-competent and responds to Hh signaling is unknown. We discussed this in (Castillo-Azofeifa et al., 2017). Although this is a really interesting problem, we argue it is beyond the scope of our manuscript since we do not introduce new data pertaining to the heterogeneous competency of lingual epithelium to generate taste buds in response to SHH. We hope the reviewers agree.

5. Additional textual changes.A. To place the problem in context for a broad readership, a schematic explaining the relationship between K14 and SHH in the developing taste bud; what is known and what your experimental data added. As written, the ideas are not well explained as a whole concept.

This point is well taken. However, the relative expression of KRT14 and SHH has not been directly reported, although SHH and KRT8 expression have been shown to largely albeit not completely overlap once taste placodes are specified (Liu et al., 2013). While there are numerous reports of KRT14 immunostaining of embryonic mouse tongues in tissue sections, there has been wide disagreement as to the actual timing of first detection of these intermediate filament proteins in developing lingual epithelium. In part, we suspect this is due to large variation in fixation protocols, as long fixation tends to block access of antibodies to KRT14 epitopes. Thus, if lowly expressed at early stages, hard fix will lead to an interpretation that KRT14 is not expressed. Thus, no studies to date allow us to build a definitive schematic of KRT14 and SHH expression. We hope the reviewers accept our attempt at clarification in the text. Lines: 32-35

References:

Castillo-Azofeifa, D., Losacco, J.T., Salcedo, E., Golden, E.J., Finger, T.E., Barlow, L.A., 2017. Sonic hedgehog from both nerves and epithelium is a key trophic factor for taste bud maintenance. Development 144, 3054-3065.

Donnelly, C.R., Gabreski, N.A., Suh, E.B., Chowdhury, M., Pierchala, B.A., 2018. Non-canonical Ret signaling augments p75-mediated cell death in developing sympathetic neurons. J Cell Biol 217, 3237-3253.

Hasegawa, H., Abbott, S., Han, B.X., Qi, Y., Wang, F., 2007. Analyzing somatosensory axon projections with the sensory neuron-specific Advillin gene. J Neurosci 27, 14404-14414.

Lu, W.J., Mann, R.K., Nguyen, A., Bi, T., Silverstein, M., Tang, J.Y., Chen, X., Beachy, P.A., 2018. Neuronal delivery of Hedgehog directs spatial patterning of taste organ regeneration. Proc Natl Acad Sci U S A 115, E200-E209.

Miura, H., Scott, J.K., Harada, S., Barlow, L.A., 2014. Sonic hedgehog-expressing basal cells are general post-mitotic precursors of functional taste receptor cells. Dev Dyn 243, 1286-1297.

Sukumaran, S.K., Lewandowski, B.C., Qin, Y., Kotha, R., Bachmanov, A.A., Margolskee, R.F., 2017. Whole transcriptome profiling of taste bud cells. Scientific reports 7, 7595.

Tang, T., Donnelly, C.R., Shah, A.A., Bradley, R.M., Mistretta, C.M., Pierchala, B.A., 2020. Cell non-autonomous requirement of p75 in the development of geniculate oral sensory neurons. Scientific reports 10, 22117.

Zurborg, S., Piszczek, A., Martinez, C., Hublitz, P., Al Banchaabouchi, M., Moreira, P., Perlas, E., Heppenstall, P.A., 2011. Generation and characterization of an Advillin-Cre driver mouse line. Mol Pain 7, 66.